# Antisense transcription-dependent chromatin signature modulates sense transcript dynamics

Thomas Brown[†] , Françoise S Howe[†], Struan C Murray[†], Meredith Wouters, Philipp Lorenz ,
Emily Seward , Scott Rata, Andrew Angel[*] & Jane Mellor[**]

## Abstract

Antisense transcription is widespread in genomes. Despite large differences in gene size and architecture, we find that yeast and human genes share a unique, antisense transcription-associated chromatin signature. We asked whether this signature is related to a biological function for antisense transcription. Using quantitative RNA-FISH, we observed changes in sense transcript distributions in nuclei and cytoplasm as antisense transcript levels were altered. To determine the mechanistic differences underlying these distributions, we developed a mathematical framework describing transcription from initiation to transcript degradation. At *GAL1*, high levels of antisense transcription alter sense transcription dynamics, reducing rates of transcript production and processing, while increasing transcript stability. This relationship with transcript stability is also observed as a genome-wide association. Establishing the antisense transcription-associated chromatin signature through disruption of the Set3C histone deacetylase activity is sufficient to similarly change these rates even in the absence of antisense transcription. Thus, antisense transcription alters sense transcription dynamics in a chromatin-dependent manner.

**Keywords** antisense transcription; chromatin; sense transcript dynamics; Set3C lysine deacetylase; stochastic model

**Subject Categories** Chromatin, Epigenetics, Genomics & Functional Genomics; Genome-Scale & Integrative Biology; Transcription

**Mol Syst Biol. (2018) 14: e8007**

## Introduction

The transcription of genomes is not limited to the transcription of genes alone. Transcription is a universally pervasive and interleaved process, with transcription events initiating from regulatory sequences such as enhancers, divergently from gene promoters, and on the antisense strand of genes (Tisseur *et al*, 2011; Lam *et al*, 2014; Mellor *et al*, 2016; Murray & Mellor, 2016). Nascent

transcripts at enhancers or around promoters can be used to recruit (Battaglia *et al*, 2017) and activate (Bose *et al*, 2017) epigenetic modifiers associated with chromatin and to activate neighbouring genes in a cell type-specific manner (Werner *et al*, 2017). This may explain why some chromatin modifications only appear after transcription has initiated (Howe *et al*, 2017). In addition to transcripts, co-transcriptional processes also influence chromatin modifications in genomes. Transcription of many non-coding transcripts uses a form of RNA polymerase II (RNAPII) that is depleted for conserved features normally associated with efficient transcription elongation including serine 2 phosphorylation of the C-terminal domain (CTD) on the largest RNAPII subunit, the H3K36 methyltransferase Set2 and the elongation factor Paf1 (Murray *et al*, 2015; Fischl *et al*, 2017). One such class of non-coding transcripts contains the nascent transcripts transcribed from the antisense strand of the gene, which are often rapidly degraded by exonucleases (He *et al*, 2008; Neil *et al*, 2009; van Dijk *et al*, 2011). Although antisense transcription within genes is a consistent feature of eukaryotic genomes (Mellor *et al*, 2016), it is not known whether it is simply a by-product of gene transcription, whether there are consequences of antisense transcription and, if so, whether these are conserved across species. Much effort has been expended to determine the function(s) associated with antisense transcription. For a small number of yeast genes, sense and antisense transcription appear to suppress one another and/or be reciprocally regulated (Hongay *et al*, 2006; Camblong *et al*, 2007; Houseley *et al*, 2008; Castelnuovo *et al*, 2013). However, there is no obvious global relationship between sense and antisense transcription, as levels at the same gene are not correlated genome-wide, either positively or negatively (Murray *et al*, 2015) and a recent study found that, at the protein level, gene expression is unaffected by lowering levels of antisense transcription in the majority of the 162 genes studied (Huber *et al*, 2016).

As antisense transcription often proceeds into the sense promoter of its associated gene (Xu *et al*, 2011; Mayer *et al*, 2015) and does not appear to be contemporaneous with sense transcription (Castelnuovo *et al*, 2013; Nguyen *et al*, 2014), we previously hypothesized that antisense-transcribing RNAPII might indirectly influence sense transcription by modulating the chromatin environment in the vicinity of the sense promoter. Thus, one round of

Department of Biochemistry, University of Oxford, Oxford, UK
*Corresponding author. Tel: +44 1865 613325; E-mail: andrew.angel@bioch.ox.ac.uk
**Corresponding author. Tel: +44 1865 613241; E-mail: jane.mellor@bioch.ox.ac.uk
[†]These authors contributed equally to this work

antisense transcription would be sufficient to leave an epigenetic signature and influence sense transcription. We identified in yeast a chromatin signature at the sense promoter and in the early coding region unique to genes with high levels of antisense transcription: high levels of nucleosome occupancy leading to a reduced nucleosome-depleted region (NDR), high histone H3 lysine acetylation and histone turnover, but low levels of histone H3 lysine 36 tri-methylation (H3K36me3), H3K79me3 and H2BK123 mono-ubiquitination, amongst others (Murray *et al*, 2015). Some of these features have been found associated with antisense transcription in mammals (Lavender *et al*, 2016). Conservation of chromatin features associated with antisense transcription between yeast and mammals will enable us to apply anything we learn in yeast about the mechanistic consequences of antisense transcription more broadly. Here we address the question of how antisense transcription influences sense transcription using a stochastic model of transcription and quantitative data from a single-molecule approach, RNA fluorescence *in situ* hybridization (RNA-FISH). This allows for the best understanding of the effects of antisense transcription on the dynamics of sense transcript production and processing at the individual cell level. We model RNA-FISH data obtained from engineered constructs expressing high or low levels of antisense transcription, but the same level of sense transcripts, thus mimicking the commonly reported situation where antisense transcription has little effect on steady-state transcript levels. We show that antisense transcription decreases rates of transcript production and processing while increasing transcript stability and, importantly, that these changes in transcription dynamics are directly influenced by the antisense-dependent chromatin signature. As we reveal a remarkably conserved chromatin architecture around the sense promoter and early transcribed region of yeast and human genes with antisense transcription, despite large differences in gene size, we suggest that the effect of antisense transcription is likely to be conserved between yeast and human genes.

# Results

### A conserved arrangement of sense and antisense transcription start sites in yeast and human genes

To address whether and how antisense transcription is conserved across species, it was necessary to map genic transcription start sites (TSSs) as either *sense* sites (sTSS), or *antisense* sites (asTSS), depending on their orientation relative to their proximal gene, and the extent of sense and antisense transcription downstream of these TSSs (Fig 1A). As many antisense transcripts are unstable, we used data from nascent transcript mapping techniques such as NET-seq (Churchman & Weissman, 2011; Nojima *et al*, 2015), PRO-seq (Booth *et al*, 2016) or GRO-seq (Core *et al*, 2008) to assess genome-wide levels of transcription in *Saccharomyces cerevisiae* and HeLa cells. To map TSSs, we used Cap Analysis of Gene Expression (CAGE) data for HeLa cells (FANTOM Consortium *et al*, 2014), pooling the polyadenylated and non-polyadenylated tag data from nuclear, cytoplasmic and whole cell fractions, and TIF-seq for yeast (Pelechano *et al*, 2013), supplemented with data from cryptic unstable transcripts (Neil *et al*, 2009) and stable unannotated transcripts (Xu *et al*, 2009). From over 20,000 protein-coding genes, we identified 9,320 with a sTSS in HeLa cells. Of these genes, we found 2,468

(27%) with an internal asTSS; 1,008 (40%) of these asTSSs were within 500 bp of the sTSS, with a median distance of 632 bp (Fig 1B). Thus, a large fraction of active genes in HeLa cells show evidence of a productive, antisense-oriented transcription start site close to their promoter. We defined 5,222 yeast genes with a sTSS, of which 1,529 (29%) had an asTSS. The median distance between the sTSSs and asTSSs of yeast genes was 884 bp, 252 bp larger than in humans (Fig 1C).

Strikingly, in humans, the asTSS aligned more closely to the 1st exon–intron boundary than the sTSS, and with a much higher frequency than expected if asTSSs are randomly re-distributed over this region (Fig 1D). In fact, in 2,162 (88%) genes with antisense transcripts, the asTSS is closer to the 1st exon–intron boundary than it is to the sTSS. Using the same approach, we also found that the asTSS aligned much more tightly with the 1st exon–intron boundary than it did with the 2nd exon–intron boundary (Fig 1E) or to the 3′ end (Fig 1F). In yeast, antisense transcription tends to initiate from the vicinity of the 3′ region of genes (Fig 1G; Xu *et al*, 2011) rather than from introns (Fig 1H); 1,202 (79%) genes with an antisense transcript had an asTSS closer to their 3′ end than to their sTSS (Fig 1G and I). Despite their distinct sites of origin in humans and yeast (1st intron–exon boundary and 3′ end, respectively; Fig 1D, G, and I) and gene size (Fig 1J), the asTSS is at a similar median distance to the sTSS in humans and yeast (632 bp compared to 884 bp), suggesting a conserved arrangement.

### Higher nucleosome occupancy at promoters of genes with high antisense transcription in yeast and humans

To examine how antisense transcription influences the chromatin and sense transcription in the vicinity of the promoter, we assessed three regions: 300 nucleotides upstream of the sTSS (the sense promoter), 300 nucleotides downstream of the asTSS (the antisense promoter) and the region between the two TSSs, which was broken into an equal number of bins. We compared the upper and lower quintiles of genes with an asTSS, giving us two groups of 494 genes in humans, and 306 and 307 genes in budding yeast. Firstly, we compared levels and distributions of sense and antisense transcription in the three regions using NET-seq, GRO-seq or PRO-seq, which are similar when comparing HeLa cells with yeast and the different techniques (Figs 2A, and EV1A and B). From this point, in Figs 2B–E and 3, data are related to profiles for NET-seq, while the PRO-seq and GRO-seq profiles are in Figs EV1 (relates to Fig 2) and EV2 (relates to Fig 3). Next, we examined a genome-wide map of nucleosome occupancy (MNase-seq). Both species had a NDR at the asTSS and a marked *increase* in nucleosome occupancy in the vicinity of the sTSS in genes with the highest levels of antisense transcription (Figs 2B, and EV1C and D). This suggests that antisense transcription may modulate promoter chromatin in both species *without* necessarily altering levels of sense transcription in the vicinity of the sense promoter (Figs 2C and D, and EV1E, F, and G). Indeed, nucleosome occupancy and sense transcription may well be disconnected (Nocetti and Whitehouse, 2016), despite the apparent association between sense transcription and nucleosome depletion at the promoter (Fig 2B). Nucleosomes also interact with one another in local space, one component of the chromatin conformation in the nucleus (Hsieh *et al*, 2015). To see whether the antisense-associated increase in nucleosome occupancy is related to the

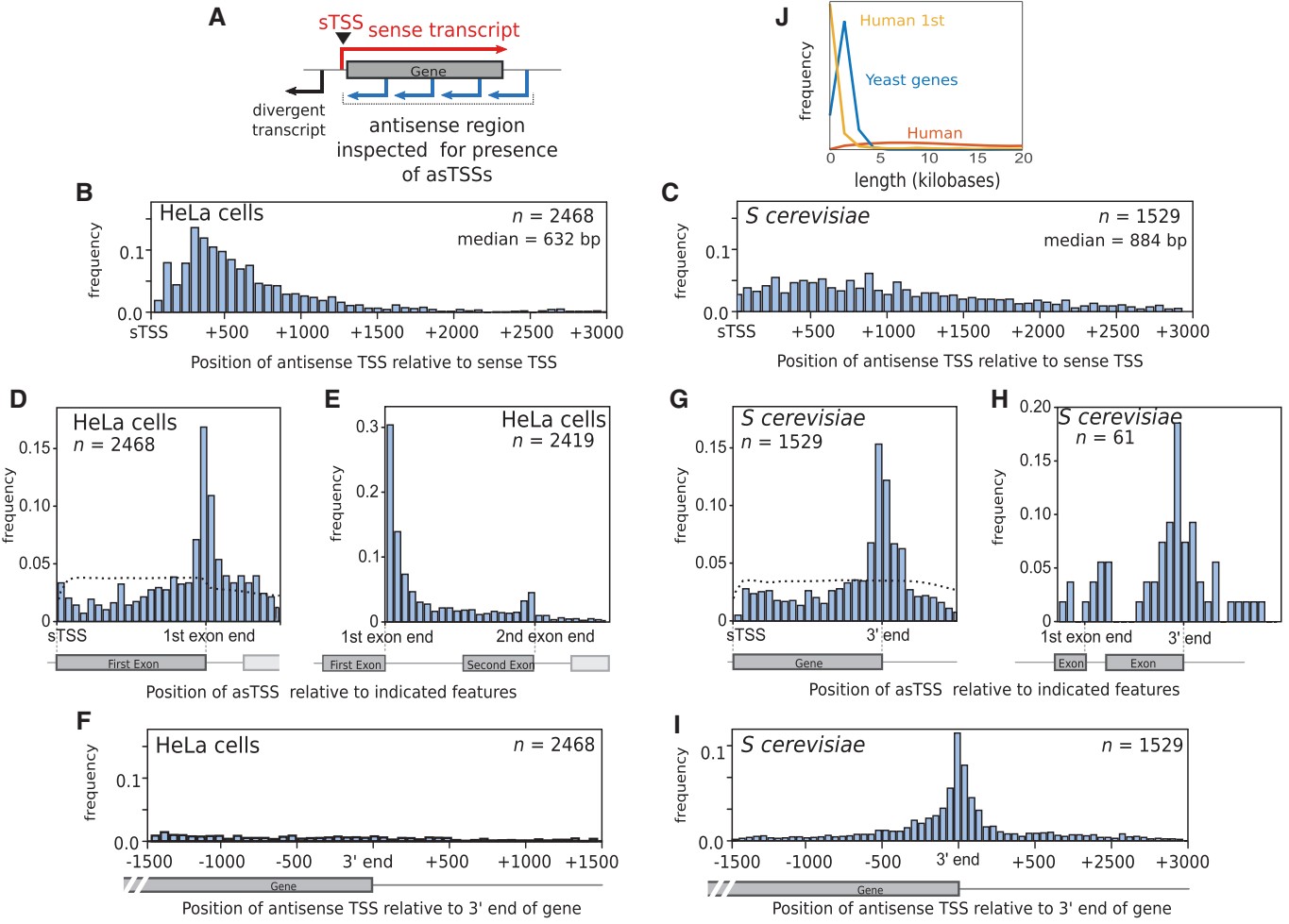

**Figure 1. Antisense transcripts initiate at a similar distance from the sense TSS in yeast and humans, though from a distinct functional site.**

A   A schematic demonstrating how antisense TSSs (asTSSs) were defined in this study. Blue arrows represent possible sites of antisense transcript initiation, within the region inspected for the presence of asTSSs, as defined by CAGE.

B   The distribution of distances between sTSS and asTSS in HeLa cells, for those 2,468 genes which had both an upstream sTSS and an internal asTSS defined by CAGE. Shown is the median distance between the sTSS and asTSS.

C   The distribution of distances between the sTSS and asTSS in *Saccharomyces cerevisiae* (budding yeast), for those 1,529 genes that had both an overlapping sense and antisense transcript, defined by TIF-seq.

D   The position of the asTSS relative to the sTSS *and* the end of the 1st exon in HeLa cells, to demonstrate which of the two points the asTSS aligns to preferentially. This position was defined as the distance between the sTSS and the asTSS, divided by the distance between the sTSS and the end of the 1st exon. The genes are the same as those shown in (B). The dotted line represents the average distribution from a thousand simulations, in which asTSSs for each gene were randomly reassigned to a base pair within the region shown.

E   The position of the asTSS relative to the end of the 1st exon and the end of the 2nd exon, for those HeLa genes in (D) that also had a second exon.

F   The distribution of distances between the 3′ end of genes and asTSS in HeLa cells, for the same genes in (B).

G   The position of the asTSS relative to the sTSS and the 3′ end of the open reading frame of the 1,529 *S. cerevisiae* genes that have both an overlapping sense and antisense transcript. The dotted line was generated as in (D).

H   The position of the asTSS relative to the end of the 1st exon and the 3′ end of the open reading frame of those *S. cerevisiae* genes in (G) that also have an intron.

I   The distribution of distances between the 3′ end of *S. cerevisiae* genes and asTSS in, for the same genes in (C).

J   The distribution of lengths for human 1st exons, human genes, and yeast genes.

chromatin conformation, we used Micro-C data in yeast, which identifies chromosomal contacts at the resolution of nucleosomes (Hsieh *et al*, 2015). For the 5,222 yeast genes that had an associated sense transcript as described above, we determined the level of gene *compaction*, as defined by Hsieh *et al* (2015), normalized such that it is independent of gene length. Strikingly, we found an inverse association between intragenic contacts and antisense transcription.

Genes with an antisense transcript showed a significantly reduced level of gene compaction, regardless of the level of sense transcription ($P = 1.2 \times 10^{-65}$, $P = 2.2 \times 10^{-39}$, Wilcoxon rank sum test, Fig 2E), suggesting that antisense transcription favours a looser higher order structure. Varying sense transcription results in *no* significant change in the level of compaction ($P = 0.29$, $0.42$, Wilcoxon rank sum test, Fig 2E).

**Antisense transcription is associated with a similar unique chromatin signature in yeast and humans**

We next turned our attention towards histone modifications (Fig 3, NET-seq; Fig EV2, PRO-seq and GRO-seq; Appendix Fig S1). Levels of H3K36me3 and H3K79me3 were higher in the region bounded by the sTSS and asTSS for those genes with high *sense* transcription, in both species (Fig 3A; Ng *et al*, 2003; Pokholok *et al*, 2005). Strikingly, however, these two modifications are much lower in those genes with high levels of *antisense* transcription, in both humans and yeast (Fig 3A). This is despite the fact that the level of sense transcription is the same in the high/low antisense classes (see Figs 2C and D, and EV1E, F, and G). That these modifications should have reverse associations with sense and antisense in *both* yeast and humans is intriguing, and suggests there may be some fundamental difference to the two modes of transcription that is shared across species. By contrast, levels of H3K4me3 tended to be more evenly spread between the sTSS and the asTSS in genes with high antisense transcription compared to high sense transcription (Fig 3B). Levels of H3K4 lysine monomethylation (H3K4me1) tended to be lower with high sense or antisense transcription showing that not all modifications have reciprocal patterns with sense or antisense transcription. Finally, levels of H3 acetylation are increased in the presence of antisense transcription (Fig 3C), particularly in the region downstream of the sense TSSs.

Taken all together, one can see that despite the vast differences in size between yeast and human genes, they share a very similar arrangement in terms of where their antisense transcripts initiate relative to their coding-transcript start site, and in how antisense transcription associates with numerous shared chromatin features. We conclude that antisense transcription in the vicinity of the sense promoter is associated with increased histone lysine acetylation and nucleosome occupancy, and decreased histone H3K36me3, H3K79me3 and chromatin compaction, and that this unique architecture is conserved between yeast and humans.

Is there a consequence of this widespread and conserved antisense transcription initiating downstream from the sense promoter for the genes in yeast and humans that have it? How might it be changing gene behaviour? To address this, we developed a mathematical model that describes the dynamics of transcription and allows us to discriminate between transcriptional events at the sense promoter, the nucleus and the cytoplasm. When compared with experimental data, the model allows us to determine which parameters of sense transcription production and processing are affected by antisense transcription.

**A stochastic model for transcription**

Our stochastic model of transcription captures the production, processing and destruction of a transcript (Fig 4A) and builds on existing models of transcription (Raj *et al*, 2006; Zenklusen *et al*, 2008; Choubey *et al*, 2015). Within the model, a gene promoter is allowed to switch between an active and inactive state stochastically, with an activation rate $\alpha$ and an inactivation rate $\beta$. In the active state, transcription initiation occurs with rate $\gamma$. As a result, the "mean production rate", that is the average rate of transcript initiation, is given by $\alpha\gamma/(\alpha + \beta)$.

Nuclear transcript processing, meanwhile, is modelled as a sum of reactions representing the advancement of RNAPII across the DNA as a series of stochastic jumps. The time to fully process a nuclear transcript is distributed as a sum of $N$ exponentials, corresponding to gene length, with parameter $k$ corresponding to elongation rate, $\Gamma(N, k)$. Our experimental protocol does not allow for the separation of nascent and nuclear transcripts; therefore, in our modelling framework, we do not differentiate between the two types of transcripts. As a result, the parameter $k$ is a conflation of both elongation rate and nuclear export rate. We refer to this parameter as the "nuclear processing rate", representing the time for a transcript to go from initiation to export. Finally, transcripts are assumed to degrade in the cytoplasm with a constant half-life, decaying exponentially with degradation rate $\delta$.

We used this model to study the transcription dynamics of the inducible yeast gene *GAL1* (Fig 4B). The first strain (high AS) contains an engineered form of the *GAL1* gene that expresses a stable antisense transcript as a result of insertion of the *ADH1* transcription terminator (*GAL1::ADH1*t; Murray *et al*, 2012, 2015). In the second strain (low AS), a 6-bp AT rich sequence within the inserted terminator region is scrambled (while retaining the overall base composition), resulting in a significant reduction in levels of the antisense transcript (Fig 4C), which is a consequence of reduced levels of antisense transcription (Murray *et al*, 2015), but no change in levels of sense transcripts (Fig 4D).

Two types of data were used to parameterize the dynamics of transcription. Firstly, we obtained the rate of degradation of cytoplasmic sense transcripts (Fig 4E), relying on the galactose-inducible and glucose-repressible nature of *GAL1*. We grew cells in galactose-containing media, then recorded the decreasing concentration of sense mRNA via Northern blot at multiple timepoints after cells were moved to glucose-containing media. By fitting an exponential curve to the data sets, we obtained the degradation rate $\delta$, which gives the half-life via the formula: $t_{1/2} = \ln 2/\delta$.

Secondly, we obtained the distribution of individual sense transcripts in the nucleus and cytoplasm of cells using RNA-FISH (Fig 4F). We probed cells grown in galactose-containing media for 2 h for the *GAL1* sense transcript and counted the number of fluorescent foci within the nucleus and cytoplasm. Individual dots were assumed to represent at least one transcript, with the number of transcripts at a given dot determined by dividing the intensity of the dot by the median intensity of all foci. Several hundreds of cells were considered for a given experiment, and the distributions of nuclear and cytoplasmic transcript counts were obtained (Fig 4G). The nuclear distribution gives an indication of mean initiation rate relative to nuclear processing rate, or the fraction $\alpha\gamma/(\alpha + \beta)k$. The cytoplasmic distribution, correspondingly, tells us the mean initiation rate relative to degradation rate, or $\alpha\gamma/(\alpha + \beta)\delta$.

Using the measured degradation rate $\delta$, we find the parameters that best fit the RNA-FISH data via the Kolmogorov–Smirnov test, sampling 1,000,000 parameter sets via Latin Hypercube (McKay *et al*, 1979) and sampling the 1,000 parameter sets with the best Kolmogorov–Smirnov statistic. The parameters obtained by simulation are then used to determine the mean initiation rate $\alpha\gamma/(\alpha + \beta)$ and nuclear processing rate $k$. The 1,000 best parameter sets obtained from the cytoplasmic data give a probability distribution for the expected true value of the parameter. The nuclear data show the likely corresponding ratio of mean initiation rate to nuclear

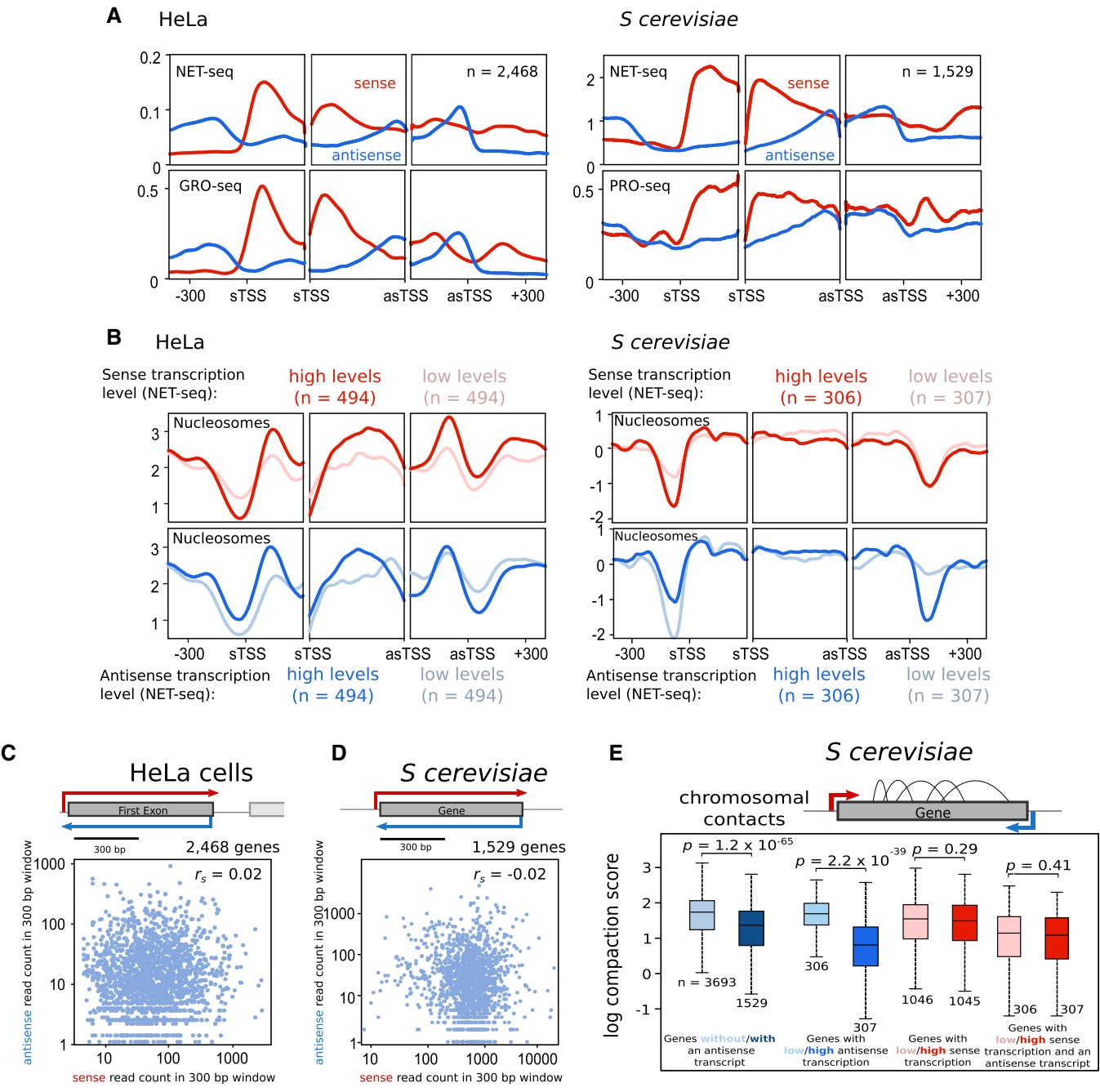

**Figure 2.   Antisense transcription is associated with changes in chromatin structure in both yeast and humans, but not with changes to the level of sense transcription.**

A     The average levels of sense and antisense transcription determined for both HeLa and *Saccharomyces cerevisiae* using NET-seq, HeLa using GRO-seq and
      *S. cerevisiae* using PRO-seq. For each trio of panels, the left panel shows average levels around the sTSS, the right panel shows the average levels around the asTSS,
      and the middle panel shows the average level within thirty equal sized bins within the region bound by the sTSS and asTSS. In all cases, levels of transcription on
      the sense strand are shown in red, while levels on the antisense strand are shown in blue. Genes considered are those which contained an asTSS, as defined in Fig 1.
B     The average levels of nucleosome occupancy determined for both HeLa and *S. cerevisiae* using MNase-seq. Panels are grouped in threes and show average levels as
      in (A). The top panels compare two sets of genes—those with high levels of sense transcription (dark red), and those with low levels (pale red). The bottom panels
      show those genes with high levels of *antisense* transcription (dark blue), and those with low levels (pale blue).
C, D   Scatter plots comparing the number of sense and antisense NET-seq reads within the 300 bp window shown, in both HeLa cells and *S. cerevisiae*, for those genes
      with both an sTSS and asTSS. Shown for both species is the Spearman correlation coefficient, $r_s$.
E     Boxplots showing the distribution of gene compaction in different sets of *S. cerevisiae* genes. Gene compaction was determined by summing the number of
      intragenic contacts, measured by Micro-C, and dividing by gene length. On each boxplot, the central mark indicates the median, and the bottom and top edges of
      the box indicate the 25th and 75th percentiles respectively. The whiskers extend to the most extreme data points. The numbers at the bottom of each box plot show
      the number of genes in that group.

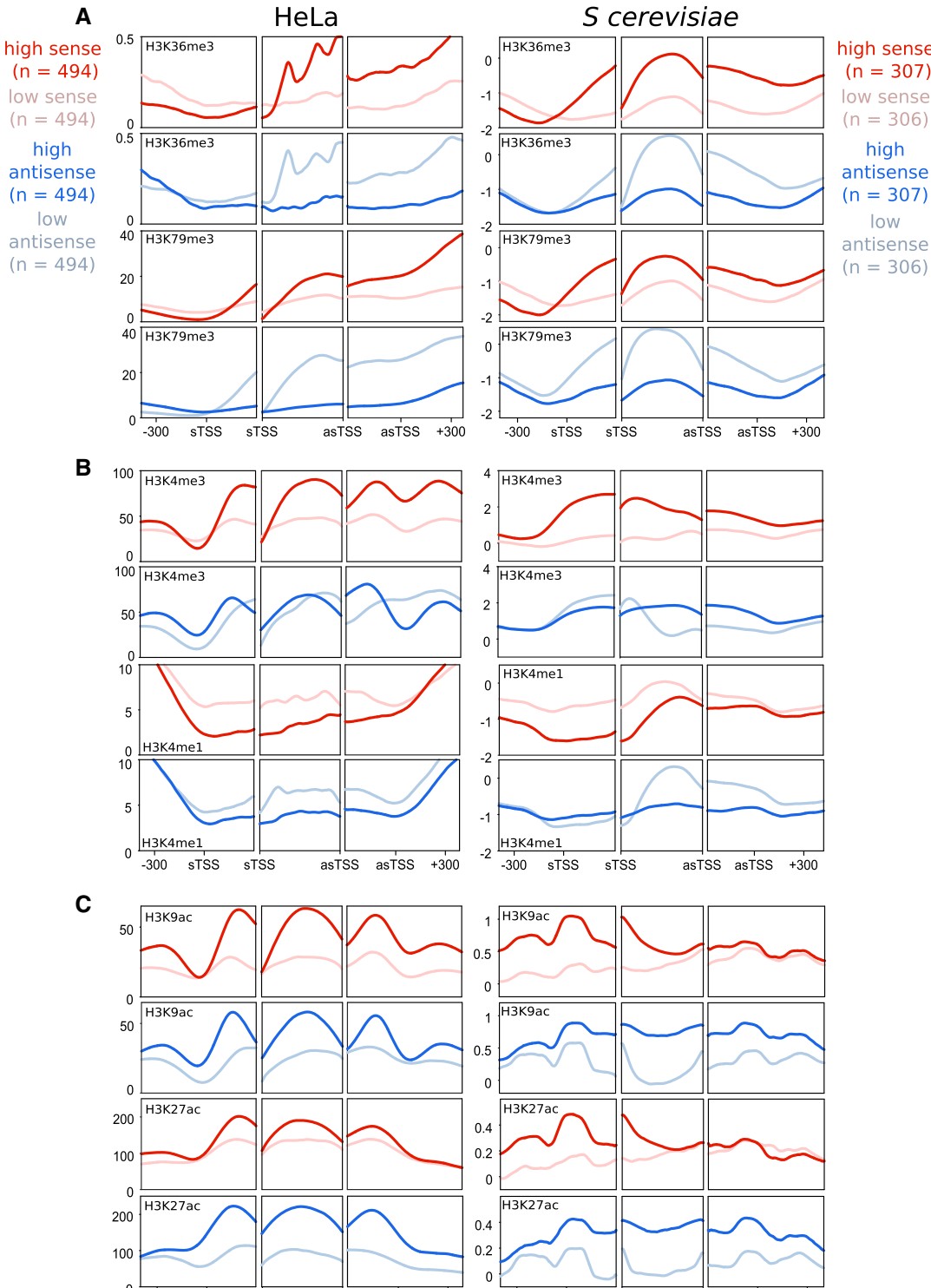

**Figure 3.   Antisense transcription has similar associations with chromatin modifications in both yeast and humans.**

A   The average levels of H3K36me3 and H3K79me3 relative to levels of sense or antisense transcription in HeLa and *Saccharomyces cerevisiae* genes. For each trio of panels, the left panel shows average levels around the sTSS, the right panel shows the average levels around the asTSS, and the middle panel shows the average level within thirty equal sized bins within the region bound by the sTSS and asTSS. Genes considered are selected from those which contained an asTSS, as defined in Fig 1. Shown in red are two sets of genes—those with high levels of sense transcription (dark red), and those with low levels (pale red). Shown in blue are those genes with high levels of *antisense* transcription (dark blue), and those with low levels (pale blue).

B   Average levels of H3K4me3 and H3K4me1, laid out as in (A).

C   Average levels of H3K9ac and H3K27ac, laid out as in (A).

processing rate. By projecting the probability distribution for the mean initiation rate onto the linear relationship, we obtain a probability distribution for the nuclear processing rate (Fig 4H). Taking the most likely value from the probability distribution, we obtain the inferred parameter values for the given strain. The various steps involved in generating rates for initiation of transcription ($min^{-1}$) and nuclear processing rate ($min^{-1}$) are shown in Fig 4H. Therefore, for any strain we can obtain the mean initiation rate, nuclear processing rate and degradation rate, corresponding to promoter, nuclear and cytoplasmic effects on sense transcript dynamics (Fig 4).

## Antisense transcription influences rates of sense transcription initiation, transcript processing and sense transcript stability

We generated experimental data using the two strains in which *GAL1* was subject to different levels of antisense transcription (Fig 4B). By obtaining transcriptional parameters in these two strains, we were able to compare how antisense transcription influences sense transcription and transcripts. Strikingly, the stability of the engineered *GAL1* sense transcripts is higher with greater antisense transcription ($t_{1/2}$ = 13.53 vs. 20.26 min for low vs. high AS; Figs 4E and 5A). Modelling of the RNA-FISH data reveals roles for antisense transcription in controlling the rates of initiation and nuclear processing of *GAL1* transcription/transcripts; both parameters were lower in the construct expressing higher antisense transcription (Figs 5B, and EV3A and B). The mean production rate was 0.425 $min^{-1}$ in the construct with low antisense transcription, but 0.256 $min^{-1}$ in the construct with high antisense transcription. Similarly, the nuclear processing rate (combining elongation and export rates) was reduced from 2.33 to 1.54 $min^{-1}$ in the presence of antisense transcription. Thus, at the engineered *GAL1* gene, antisense transcription *does not* alter overall sense transcript levels (Fig 4D) but *does* alter the dynamics of sense transcript production, processing and turnover.

Next, we asked if the effect of antisense transcription on sense transcription/transcript dynamics extends to other genes. We show that the effect of antisense transcription on transcript stability is not strictly limited to the engineered *GAL1* genes used here (Figs 5C and EV3C). Using four different sources of data, we compared the stability of transcripts produced from 1,529 yeast genes with, and 3,693 without an antisense transcript (Wang *et al*, 2002; Churchman & Weissman, 2011; Miller *et al*, 2011; Geisberg *et al*, 2014). Remarkably, we observed significant increases in stability for sense transcripts produced from genes with an antisense transcript. Finally, we produced and modelled RNA-FISH data for transcript distributions (Fig EV3D) for five genes with varying levels of antisense transcription, modelled that data and expressed the elongation/export rate as a function of mean production rate to account for inherent differences in sense expression levels (Fig 5D). We observed a decrease in transcript processing rate as levels of antisense transcription increases, corroborating our observations at *GAL1*. Taken together, these data suggest compensating changes in rates of sense transcript production and sense transcript degradation as a result of antisense transcription.

We asked how antisense transcription alters sense transcript dynamics. It is possible that changes in transcript dynamics resulting from antisense transcription are a consequence of altered patterns of histone modification. To this end, we sought to assess whether experimentally modulating histone modifications could recapitulate the effects of changing antisense transcription, focusing on histone H3 lysine acetylation, as genes with high levels of antisense transcription tend to be associated with increased histone acetylation compared to genes with lower levels (see Fig 3; Murray *et al*, 2015). Final levels of histone acetylation are influenced by rates of acetylation and deacetylation. The histone deacetylase complexes Rpd3L (containing Rpd3, Pho23 and Hos2), Rpd3S (containing Rpd3 and Rco1) and Set3C (containing Set3 and Hos2) decrease overall levels of histone acetylation, control transcript dynamics at a small number of genes, but do not affect global gene expression (Pijnappel *et al*, 2001; Kim *et al*, 2012, 2016; Weinberger *et al*, 2012; Woo *et al*, 2017). We asked first which of these HDAC complexes control histone acetylation dependent on levels of antisense transcription and second whether one of these HDAC complexes, by its effect on histone acetylation, might modulate the same parameters as antisense transcription.

---

**Figure 4.   A stochastic model for transcription dynamics.**

A   Schematic of the model (see text for details).

B   The engineered *GAL1* expressing high or low antisense (AS) transcription (Murray *et al*, 2015). Purple line shows the position of the strand-specific probes used for Northern blotting. Sense transcripts are in red, antisense transcripts in blue.

C   Representative Northern blot showing levels of *GAL1* antisense transcripts (black arrowhead) in the high and low antisense strains during the transition from glucose (0 min) to galactose (GAL). RNA-FISH experiments are performed after 120 min in GAL. Samples were run on the same gel with the intervening lanes spliced out (as indicated by the black vertical line). The positions of the 25S and 18S rRNAs are represented by short black horizontal lines. Ethidium bromide-stained rRNA is the loading control.

D   Quantitation of *GAL1* sense transcripts levels as measured by Northern blotting in the high and low antisense strains, normalized to high AS levels. N = 9, error bars are SD, *P*-value shown above the bar calculated by paired *t*-test.

E   Representative Northern blot showing *GAL1* sense transcripts (black arrowhead) and *GAL10* lncRNA (asterisk) after transfer from GAL (0) to GLU for the time indicated (min). From these data, the rates of degradation of the transcripts are calculated, after normalization to the GAL timepoint. Samples were run on the same gel with the intervening lanes spliced out (as indicated by the black vertical line). The positions of the 25S and 18S rRNAs are represented by short black horizontal lines. N = 9.

F   Example of single-molecule RNA-FISH data showing two cells. DNA is stained with DAPI (blue) and single *GAL1* sense transcripts in green. A bright nuclear focus is present in the top cell, containing 2–3 nascent transcripts.

G   The frequency of nuclear and cytoplasmic transcripts for 1,193 individual cells averaged across nine experiments is determined using an automated foci recognition algorithm (blue bars). The red line shows the simulated distribution from the model in (A). Data shown are counts and fit for low AS *GAL1* mRNA.

H   Schematic showing how mean initiation rate and nuclear processing rates are obtained. The fit to nuclear RNA distribution dictates the ratio of mean initiation rate to nuclear processing rate and the cytoplasmic rate determines the ratio of mean initiation rate to degradation rate. By fitting the degradation rate to the shutdown data in (E), the probability distributions for mean initiation rate and nuclear processing rate are obtained outright.

---

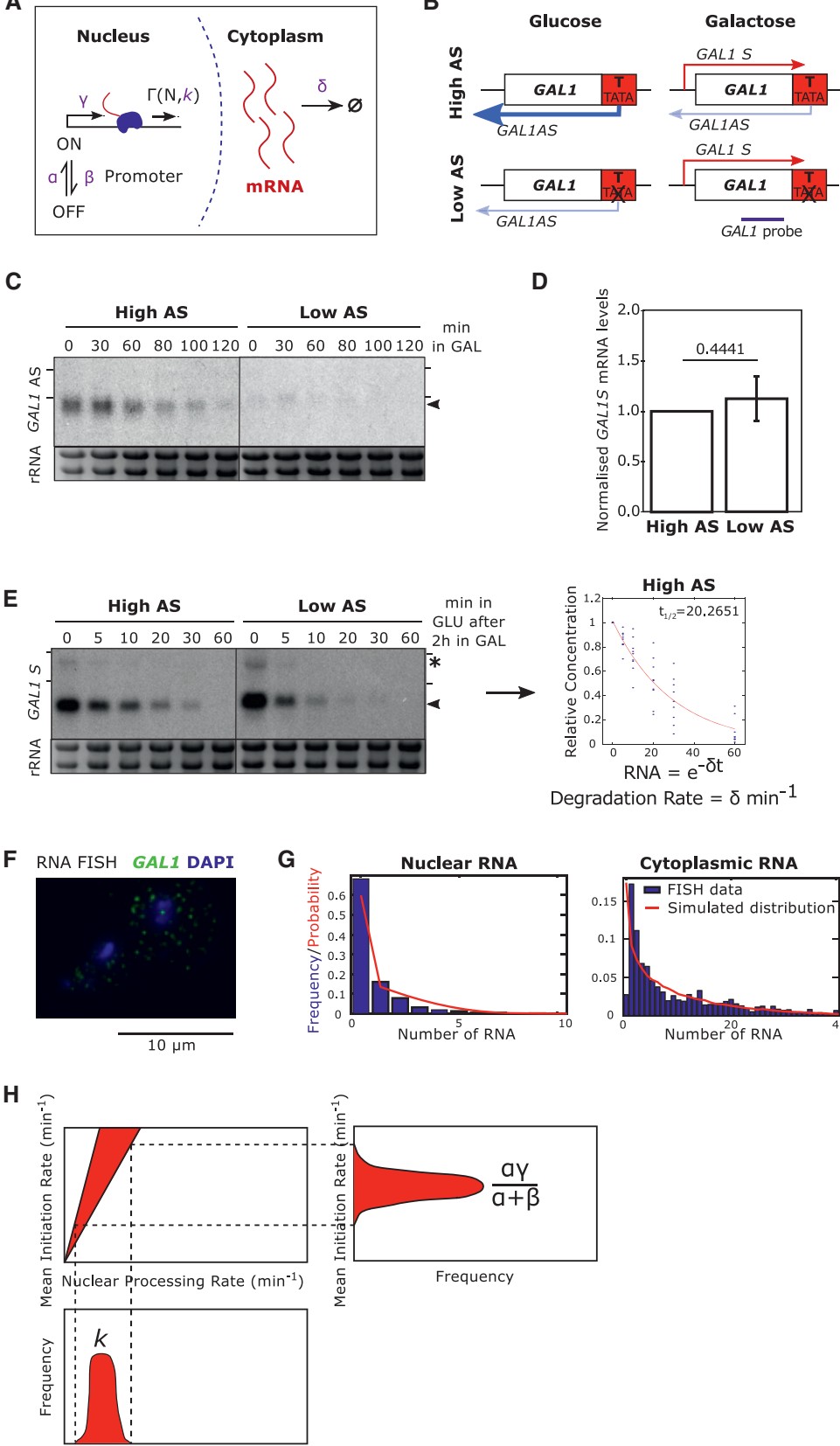

Figure 4.

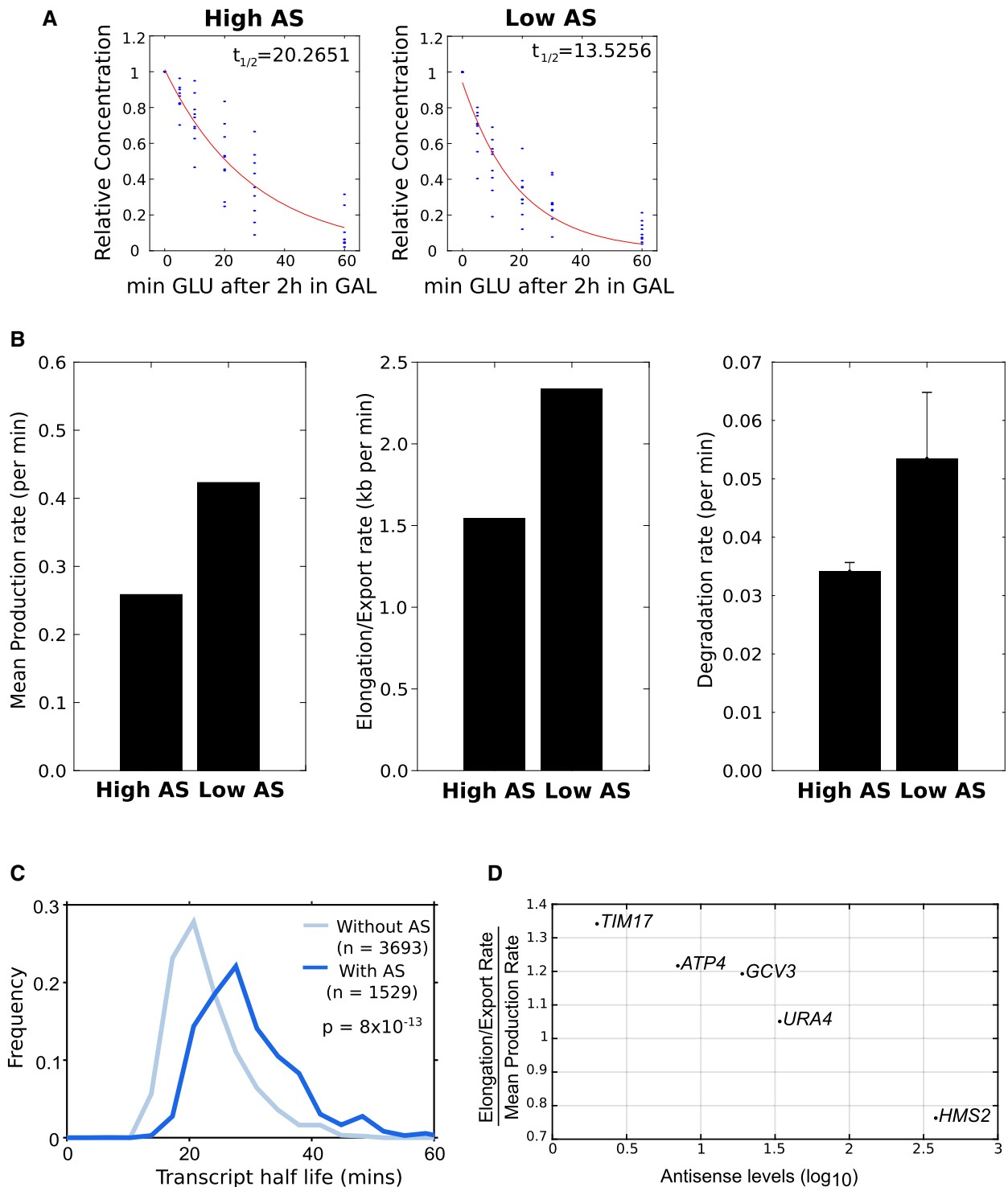

**Figure 5.  Transcription dynamics change with high antisense transcription.**

A   *GAL1* sense transcript turnover rates from the engineered *GAL1* with high (left) or low (right) antisense transcription. The red lines show the best fit for nine experiments (see Fig 4E for details).

B   Histograms showing the mean production rate (left), mean elongation/export rate (middle) and transcript degradation rate (right; error bars for degradation rates are RMSE of linear regression fit to exponential model) for the engineered *GAL1* genes expressing high or low antisense transcription.

C   Transcript stability for sense transcripts from 1,529 genes with an asTSS (dark blue) or 3,693 genes without one (light blue). The frequency plot shows the majority of transcripts have higher stability when expressed from genes with an antisense transcript.

D   Elongation/export rate expressed relative to mean production rate for five endogenous genes ranked by levels of corresponding levels of antisense transcription in a 300 bp downstream of the antisense TSS as determined by NET-seq.

### *SET3* deletion differentially influences levels of H3K9ac at genes that differ by the presence or absence of antisense transcription

Acetylation of K4, K9 and K14 on histone H3 is higher at genes with higher antisense transcription (Fig EV4A). Next, we asked whether the change in histone lysine acetylation, following gene deletion of specific HDAC components, is different when considering those genes with or without an antisense transcript. Strikingly, following deletion of *SET3* and *HOS2* (Set3C) and *RCO1* (Rpd3S), those 3,693 genes *without* an antisense transcript show a significantly larger increase in the level of acetylation than those 1,529 genes *with* an antisense ($P = 6 \times 10^{-13}$ for *set3Δ*, Figs 6A and EV4A). Furthermore, the changes were observed at different regions of the metagene, consistent with where Set3C and Rpd3S are proposed to function (Kim & Buratowski, 2009; Li *et al*, 2009). Next, we asked whether increased acetylation could be explained by increased antisense transcription in the mutant strains. This was the case for the *rco1* mutant (Murray *et al*, 2015) and so it was excluded from this study, but not for the *set3* mutant (Fig EV4B). Thus, Set3C modulates acetylation at genes with low antisense transcription suggesting some redundancy in the effect of deleting *SET3* and in the effect of antisense transcription.

### *SET3* differentially influences levels of H3K9ac at *GAL1* with high or low antisense transcription

We investigated the effect of *SET3* deletion in the presence or absence of antisense transcription at the engineered *GAL1*. Importantly, Set3C does not affect the levels of the stable antisense transcript at engineered *GAL1* (Fig 6B). That there is no change in antisense transcription is confirmed by levels of transcription-associated histone modifications H3K4me2 and H3K4me3 in the four strains, which are low without antisense transcription and higher with antisense transcription and, importantly, do not change when *SET3* is deleted (Fig EV4C).

Next, we asked how Set3C influences levels of H3K9ac at engineered *GAL1* with high or low antisense transcription (Fig 6C). High levels of acetylated H3K9 in chromatin correlate with antisense transcription (Murray *et al*, 2015). As expected, the construct with high antisense transcription has higher levels of H3K9ac than the strain with low levels of antisense transcription, despite both constructs producing similar levels of the *GAL1* sense transcript when induced (Murray *et al*, 2015; Fig 6D). On deletion of *SET3,* we observe a higher level of H3K9ac in the transcribed region in strains with low

antisense transcription (Fig 6C). This neatly reproduces what we have observed genome-wide—that *SET3* deletion has a greater effect on acetylation levels in the absence of antisense transcription.

We conclude that the effect of *SET3* deletion on levels of H3K9 acetylation at the engineered *GAL1* gene is unlikely to result from changes to antisense transcription, but from a direct effect on the chromatin. We hypothesize that antisense transcription buffers chromatin against the modulating effects of Set3C during sense transcription. Thus, following deletion of *SET3*, we would expect the transcription dynamics in the strain with low levels of antisense transcription to resemble those in the strain with high antisense transcription, assuming transcription dynamics are influenced solely by the chromatin.

### Altering levels of histone acetylation recapitulates the effect of antisense transcription on sense transcription dynamics

RNA-FISH data (Fig EV4D) and degradation rates (Fig 6E) were produced for *SET3* deletion strains expressing the engineered *GAL1* gene with either high or low levels of antisense transcription. The experimental data were then modelled to estimate the parameters of sense transcription dynamics (Figs 6F and EV4E). Consistent with our hypothesis, deletion of *SET3* in the strain with low levels of antisense transcription decreased the mean production rate, decreased the nuclear processing (elongation/export) rate and increased the stability of the mature *GAL1* sense transcripts, making the transcript dynamics of the strain with low levels of antisense transcription behave more like the strain with high antisense transcription. Global levels of transcripts are buffered by opposing rate changes for synthesis and degradation resulting in no overall change, as observed previously (Dori-Bachash *et al*, 2011). We suggest that antisense transcription reduces the sensitivity of genes to deacetylation by Set3C, and this influences transcription dynamics. Thus, at *GAL1*, antisense transcription buffers gene expression against the action of chromatin modifiers such as Set3C. We conclude that sense transcription dynamics are variable and can be modulated by histone modifiers, and therefore histone modifications, in the vicinity of the promoter and early part of the coding region.

In summary, we show that antisense transcription has a conserved spatial and chromatin architecture in both yeast and human genes, focused around the sense promoter and early transcribed region. Modelling with quantitative data reveals that antisense transcription at the engineered *GAL1* locus alters all measurable aspects of sense transcription by decreasing the rates of

---

**Figure 6.  Set3C alters transcription dynamics in an antisense-dependent manner.**

A    Strains lacking *SET3* show a larger increase in H3K9ac, relative to WT levels, at 3,693 genes without antisense transcripts compared to 1,529 genes with antisense transcripts.

B    Quantitation of *GAL1* antisense transcript levels from the high and low antisense constructs with or without *SET3*. N = 2, data points shown, error bars are SD, *P*-values are shown above the bars calculated by paired *t*-test.

C    Chromatin immunoprecipitation showing levels of H3K9ac relative to histone H3 in the strains with high or low antisense transcription in the presence or absence of *SET3*. The positions of the primer pairs for RT–qPCR are shown in the schematic below. N = 3, data points shown, error bars are SEM.

D    Quantitation of *GAL1* sense transcripts from Northern blotting of the high and low antisense constructs with or without *SET3*. N = 3 error bars are SD, *P*-values are shown above the bars calculated by paired *t*-test.

E    *GAL1* sense transcript degradation rates in *SET3* (N = 9) and *set3Δ* (N = 2) strains with high or low *GAL1* antisense transcripts.

F    Histograms showing the mean production rate (left), mean elongation/export rate (middle) and transcript degradation rate (right) for the engineered *GAL1* genes expressing high or low antisense transcription in the presence or absence of *SET3*. Error bars for degradation rates are RMSE of linear regression fit to exponential model.

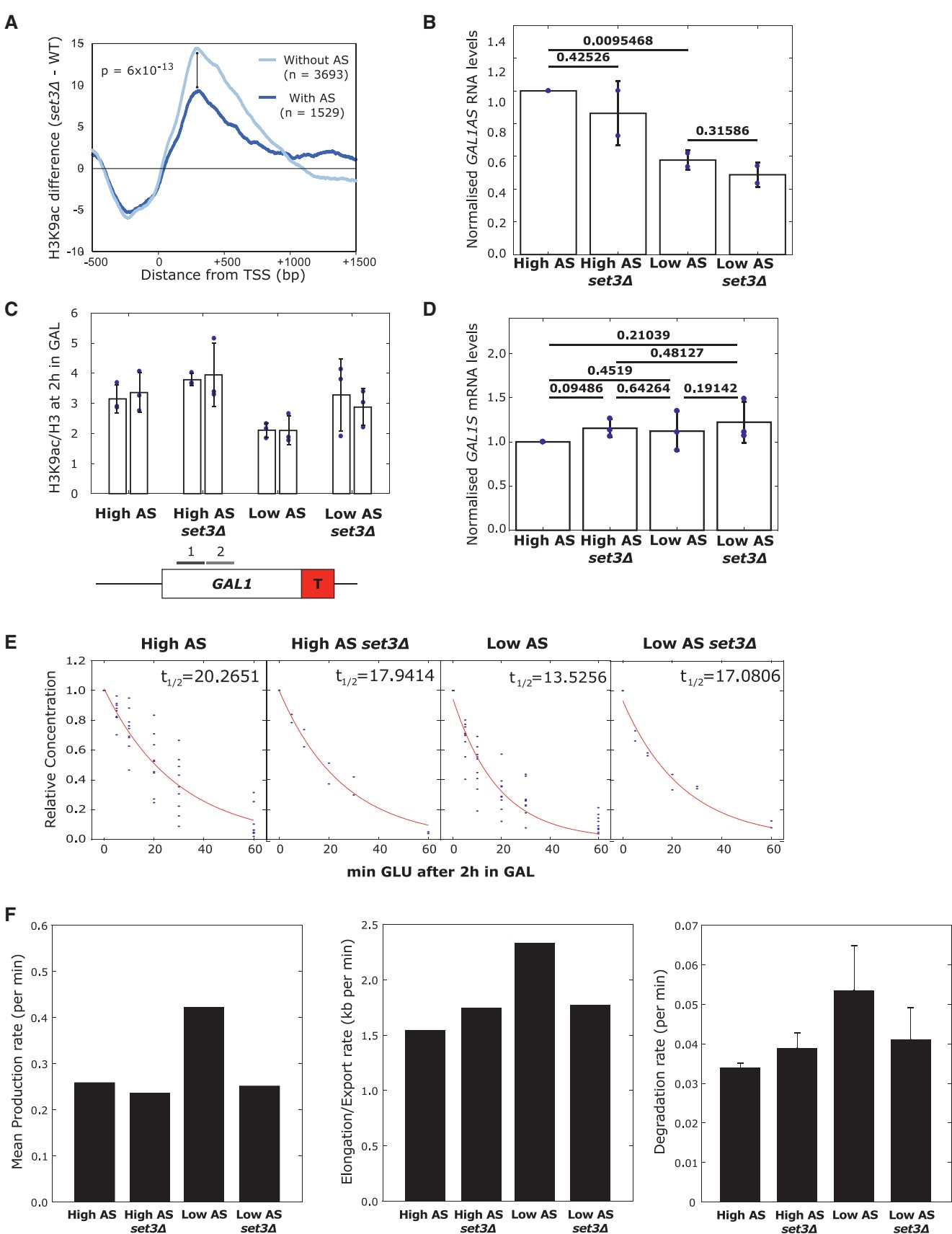

**Figure 6.**

initiation and processing of the nuclear transcripts, and the cytoplasmic degradation rate. The effect of antisense transcription on sense transcript dynamics is observed at other genes and can be mimicked at *GAL1* by simply increasing levels of histone acetylation in the vicinity of the promoter.

## Discussion

Antisense transcription is a widespread feature of both yeast and human genomes. In this work, we use mathematical modelling to provide insights into the consequences of antisense transcription on chromatin architecture and sense transcript dynamics, and show for the first time that antisense transcription alters rates of transcript production and transcript degradation.

There is a tight, apparently counterproductive, coordination between the processes of production and degradation, widely observed in yeast and mammals (Das *et al*, 2017). For example, mutants such as *rpb4Δ*, which display reduced rates of transcription, compensate by reducing the rate of transcript degradation (Schulz *et al*, 2014). The coordination of transcription and transcript degradation is known to be influenced by promoter sequences, the Rpb7 component of RNA polymerase II and many factors involved in mRNA degradation (Enssle *et al*, 1993; Dori-Bachash *et al*, 2011, 2012; Sun *et al*, 2013). Assessment of steady-state mRNA levels in mutants that influence these rates often leads to the conclusion that these factors do not have much effect on gene expression, apart from the associated stress response (O'Duibhir *et al*, 2014). This would be entirely consistent with what we observe for antisense transcription and the *set3* mutant, which do not change steady-state transcript levels at *GAL1*, but *do* change the epigenetic marks on the associated chromatin, and in doing so, alter the transcription dynamics. The *set3Δ* strain with low *GAL1* antisense transcription showed raised levels of H3K9ac at *GAL1* similar to those observed with high antisense transcription and a concomitant lowering of sense transcript production, processing and degradation rates to resemble those in strains with high antisense transcription (Fig 7).

Our model makes no assumptions about the mode of transcription initiation *ab initio* and can accommodate genes with "bursty" kinetics, as it allows for promoters to exist in both active and inactive states, or a constitutive model of production, and so would be applicable to analyse data from mammalian cells. Indeed, much transcription initiation in yeast and mammals shows bursting kinetics (Suter *et al*, 2011; Lenstra *et al*, 2016). In addition, our model extends previous models, incorporating a stochastic elongation rate, and as such, each gene displays a distribution of times from initiation to termination. One interesting finding is that our model infers a decrease in the rates of both sense transcription initiation and elongation/export in the presence of antisense transcription, yet we observe no difference in the NET-seq, GRO-seq and PRO-seq average profiles between genes with high and low levels of antisense. How could antisense transcription alter sense transcription dynamics without altering these nascent transcription profiles? These techniques offer a static, steady-state view of the locations of engaged polymerase across a gene. A corresponding decrease in initiation rate and elongation rate would result in the same number of polymerases on a gene at steady state. Any differences in polymerase profiles clearly come as a result of a number of complex interacting factors including physical blocks to transcription, the activity of elongation factors and stochastic dynamics of polymerase (Jonkers & Lis, 2015). Here we have analysed the production and processing of functional mRNA. As such, we look at the effect of antisense on the passage of the transcript through to the cytoplasm and on the rates of productive initiation, meaning that we ignore any initiation events that, for example, result in early termination. By contrast, NET-seq and GRO-seq likely convey a more complex story, in which the number of engaged polymerases does not correspond directly to the resultant number of steady-state RNAs. This is evidenced by the less-strong-than-expected Spearman correlations between nascent transcription reads and RNA-seq reads either in the 300-bp window downstream from the promoter (PRO-seq 0.52; NET-seq 0.78) or across the whole gene (PRO-seq 0.61; NET-seq 0.79).

The effects of antisense transcription on the sense promoter and promoter-proximal chromatin architecture are conserved between yeast and humans, despite large differences in gene size. In addition, antisense transcription initiates at a similar distance downstream from the sense transcription initiation site in both systems, suggesting that the effect of antisense transcription is focused on the promoter and early transcribed region of genes. This is where most control can be exerted over transcription dynamics, as the promoter and early coding region chromatin can influence initiation and the elongation phases of transcription, respectively. Moreover, we observe conserved changes to the promoter and promoter-proximal chromatin structure as the functional consequence of antisense transcription. These changes include increased lysine acetylation and increased nucleosome occupancy, which together could influence the residence time of transcription factors bound to the promoter or the composition of RNA polymerase II leaving the promoter. At the promoter itself, the altered chromatin features may also be reinforced by divergent upstream non-coding transcription (Marquardt *et al*, 2014). In support of a promoter focused function, we have recently shown that RNA polymerase II shows variable enrichment with elongation factors and that this a function of promoter sequences and associated transcription factors (Fischl *et al*, 2017). Paf1 enrichment on RNA polymerase II, for example, through its effects on the chromatin structure, affects how the encoded transcripts are decorated with RNA binding proteins that control transcript export from the nucleus.

Could antisense transcription function in gene regulation? By reducing production and increasing stability, as observed with high antisense transcription, the same final transcript response level can be achieved as with low antisense transcription, but the time taken to reach these final levels differs, and this can be a regulatory feature, for example, providing benefit in some bet-hedging strategies (Snijder & Pelkmans, 2011) or if rapidly varying conditions are expected. Indeed, antisense transcription has a proposed role in fine-tuning levels of sense transcripts under different environmental conditions (Xu *et al*, 2011) and its production can be regulated (Conley & Jordan, 2012; Murray *et al*, 2012, 2015; Nguyen *et al*, 2014). Transcription elongation is not a smooth process from start to finish, with RNA polymerase pausing heterogeneously across the gene (Jonkers & Lis, 2015), and this is likely to influence the nuclear processing rate. Changing the nuclear processing rate does not affect the cytoplasmic distribution or the rate at which a cytoplasmic steady state is reached. However, antisense transcription-dependent changes to the nuclear processing rate could be tuning how much

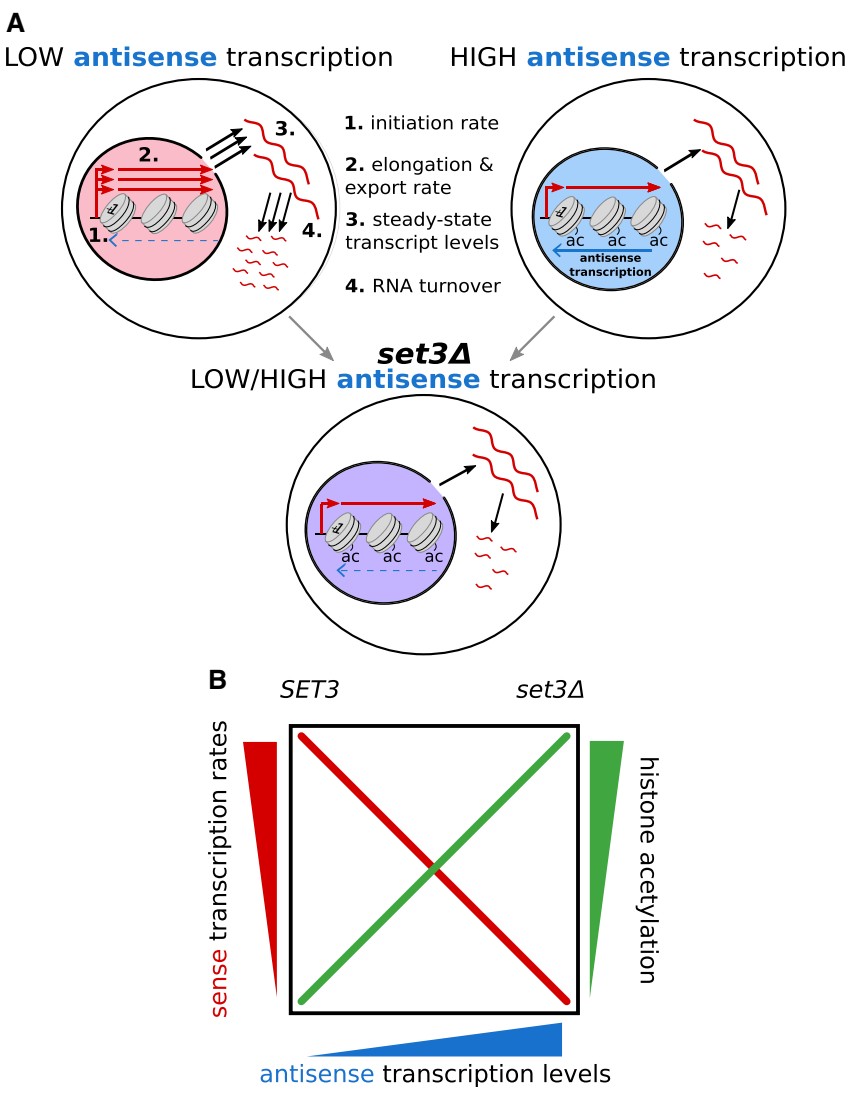

**Figure 7.  Summary.**

A  Schematic outlining the influence of antisense transcription or *SET3* deletion on levels of histone lysine acetylation (ac), steady-state transcript levels (3) and on the rates of transcript production (1), processing and export (2) and transcript decay (4). Line width indicates levels (3) or rates (1, 2, 4) with increased rate represented by extra lines.

B  Relationship of the dependent variables to independent variables in our study. High antisense transcription is associated with a highly acetylated chromatin environment, resulting in slower sense transcription kinetics, namely the initiation rate and elongation/export rate. We mimic this behaviour by deleting *SET3*, increasing acetylation levels and we observe a reduction in initiation and elongation/export rates.

other pathways can affect a transcript. For example, decreasing the elongation/export rate could make a gene more susceptible to control by a factor that relies on stochastic events during transcription or while a transcript is in the nucleus.

Why does Set3 modulate transcription dynamics predominantly at low antisense genes? We see increased levels of acetylation at low antisense genes when the integrity of Set3C is lost. Set3C may antagonize the action of the SAGA lysine acetyltransferase, as genes with high antisense show significantly more enrichment for the Spt3 component of the SAGA lysine acetyltransferase (KAT) complex (Murray *et al*, 2015), supporting a higher inherent level of acetylation at high antisense genes in strains with Set3C. This may be a reinforcing mechanism whereby once the levels of

acetylation on chromatin are high, more KAT will be recruited to maintain these levels. A dynamic interplay between Set3C and the KATs, modulated by differences in histone turnover and chromatin compaction, could explain the different characteristics of high and low antisense genes and the particular sensitivity of low antisense genes to loss of Set3C. Being able to modulate transcription and transcript dynamics by manipulating the activity of a chromatin modifying enzyme strongly supports antisense transcription modulating the chromatin structure in the vicinity of promoters and this in turn affecting transcription and transcript fate. Pervasive transcription is not limited to the antisense strand of genes as studied here, but is also abundant at enhancer elements, at the 3′ ends of genes and throughout gene-rich

regions of many different genomes including plants, yeast, fungi, flies and worms (Ni et al, 2010; Kwak et al, 2013; Andersson et al, 2014; Nojima et al, 2015; Booth et al, 2016; Krzyczmonik et al, 2017; Ietswaart et al, 2017). In some cell types, non-coding transcripts could facilitate RNAi-mediated degradation of the sense transcripts which might make uncovering the associations we have found in budding yeast, which lacks an RNAi system, more challenging. However, the functional consequences of most non-coding transcription are likely to be a result of the act of transcription, and its effects on associated chromatin, rather than via the transcript itself. In yeast, many non-coding transcription events are performed by a distinct RNA polymerase II complex, depleted for Paf1 and Set2 (leading to reduced Ser2 phosphorylation on the Pol2 CTD and reduced H3K36 methylation) amongst other factors (Fischl et al, 2017), explaining in part the unique chromatin environment associated with these events. Whether enhancers also contain a chromatin environment, for example high H3K27ac, dictated by non-coding transcription and how this influences enhancer function are questions for the future, but underscore the potential regulatory nature of non-coding transcription, as distinct from the encoded transcripts. Indeed, it is becoming clear that levels of pervasive transcription can be regulated in genomes (Mellor et al, 2016).

## Materials and Methods

Jane Mellor is responsible for all reagent and resource requests. Please contact Jane Mellor at jane.mellor@bioch.ox.ac.uk with requests and inquiries and see Appendix Table S4 for reagent details and Appendix Table S5 for details concerning software and algorithms.

### Yeast culture and genetic manipulation

Strains were streaked from glycerol stocks onto 2% agar YPD (1% yeast extract (Difco), 1% bactopeptone, 2% glucose) plates and grown (1–2 days, 30°C). Cell pre-cultures were then grown overnight in 5 ml YPD at 30°C. This culture was used to inoculate an appropriate volume of YPD culture at $OD_{600}$ 0.2 which was grown at 30°C, shaking at 200 rpm to $OD_{600}$ 0.45–0.5. To induce the GAL1 gene, cell cultures were centrifuged (900 g, 3 min) and then re-suspended in YPG (1% yeast extract (Difco), 1% bactopeptone, 2% galactose) pre-warmed to 30°C. Re-suspended cells were incubated (30°C, 200 rpm) for the specified time(s) before harvesting by centrifugation (900 g, 4 min). For the experiments to obtain the GAL1 sense degradation rates, after 2 h in YPG, cells were transferred back to fresh YPD pre-warmed to 30°C and 15 ml of samples was harvested at 0, 5, 10, 20, 30 and 60 min. Other experiments were done at $OD_{600}$ 0.6–0.8.

Genetic manipulation of strains was performed using the homologous recombination method (Longtine et al, 1998). For gene deletion strains, PCR products were made containing the HISMX or KANMX selection cassettes flanked at both ends by 40 bp of sequence homologous to sequences either side of the region to be deleted. Construction of the GAL1:ADH1t (high AS) and TATA mutant (low AS) strains has been described previously (Murray et al, 2012, 2015). Cells to be transformed were grown to log phase, pelleted, re-suspended in 450 µl 100 mM LiAc/TE and incubated (> 1 h, 4°C). 100 µl of cell suspension, 10 µl of PCR product, 10 µl calf thymus DNA (Sigma D8661) and 700 µl 40% polyethylene glycol in 100 mM LiAc/TE were incubated (30 min, 30°C) then heat-shocked (20 min, 42°C). Cells were pelleted (5 min, 4,600 g), re-suspended in $H_2O$ and plated onto appropriate selection media. DNA was extracted from the resulting colonies, screened by PCR and confirmed by sequencing.

### Chromatin immunoprecipitation (ChIP)

Yeast grown to $OD_{600}$ 0.5 in 50 ml of YPD was transferred to YPG for 2 h before they were fixed in 1% formaldehyde in 45 ml PBS for 30 min at 22°C followed by addition of 125 mM glycine for 5 min. Cell pellets were collected by centrifugation (900 g, 4 min) before washing twice with 10 ml cold PBS. Cells were re-suspended in 500 µl cold FA-150 buffer (10 mM HEPES pH 7.9, 150 mM NaCl, 0.1% SDS, 0.1% sodium deoxycholate, 1% Triton X-100) and broken using 1 ml glass beads on a MagnaLyser (Roche; 2 × 1 min runs, 2,500 g, 4°C). Sample volume was increased to 2 ml with FA-150 buffer before shearing of the fixed chromatin by sonication using a biorupter (Diagenode, 30 min, 1 min on, 20 s off, medium setting). Chromatin was cleared by centrifugation (9,400 g, 15 min, 4°C), and 50 µl was diluted to 200 µl with FA-150 buffer and incubated with 5 µl of the following antibodies as appropriate: H3, H3K4me2, H3K4me3, H3K9ac (for details see the Appendix Table S4) in 1.5 ml siliconized Eppendorf tubes for 15–20 h rotating at 4°C. Bound chromatin was immunoprecipitated for 90 min at 22°C with 50 µl protein A-Sepharose pre-blocked with bovine serum albumin and sonicated salmon sperm DNA. Beads and attached chromatin were pelleted by centrifugation (640 g, 1 min) and washed with TSE-150 buffer (20 mM Tris–Cl pH 8.0, 150 mM NaCl, 2 mM EDTA, 0.1% SDS, 1% Triton X-100) for 3 min, TSE-500 buffer (20 mM Tris-Cl pH 8.0, 500 mM NaCl, 2 mM EDTA, 0.1% SDS, 1% Triton X-100) for 3 min, LiCl buffer (0.25 M LiCl, 10 mM Tris-Cl pH 8.0, 1 mM EDTA, 1% deoxycholate, 1% NP-40) for 15 min and twice with TE. After washing, chromatin was eluted from the beads for 30 min at 65°C with elution buffer (0.1 M NaHCO₃, 1% SDS). Addition of 350 mM NaCl and incubation for 3 h at 65°C reversed the cross-links before treatment of samples with RNase A for 1 h at 37°C and proteinase K overnight at 65°C. DNA was purified using a PCR purification kit (Qiagen) and eluted in 400 µl 1 mM Tris–Cl pH 8.0. Input DNA was diluted accordingly. Real-time quantitative PCR (qPCR) was performed using a Corbett Rotorgene and Sybr green mix (Bioline). Data [(IP—no antibody control)/input] were expressed as a percentage of the input and normalized to levels of H3 where appropriate. The primers used are listed in Appendix Table S1. All ChIP experiments were performed ≥ 2 times with independent biological samples.

### RNA extraction

Fifteen millilitres of log phase yeast culture at a density of $OD_{600}$ 0.6–0.8 grown in the appropriate medium was pelleted (900 g, 3 min), re-suspended in 400 µl TES (100 mM Tris–HCl (pH 7.5), 100 mM EDTA (pH 8.0), 0.5% SDS) and 400 µl phenol:chloroform (pH 4.7) and incubated (65°C, 20 min, 22 g). The mixture was incubated (−80°C, 30 min). After spinning (15,900 g, 20 min, 4°C), the upper layer was transferred to 10 mM NaOAc pH 5.5/ethanol and incubated (−80°C, > 30 min). RNA precipitate was pelleted

(15,900 $g$, 20 min, 4°C) and re-suspended in 100 µl $H_2O$. RNA concentration was measured using a Nanodrop, and samples were diluted to 1,000 ng µl$^{-1}$.

## Northern blotting

20 µg of RNA was separated on 1.1% formaldehyde FA gels for 3 h and transferred to Hybond-N+ nylon membranes (Amersham) by wet blotting overnight in 20× SSC. After fixing the RNA to the membrane (2 h, 80°C), the membranes were blocked in PerfectHyb Plus (Sigma H7033; > 2 h, 65°C). Radio-labelled strand-specific *GAL1* sense and *GAL1* antisense probes were generated using asymmetric PCR with the primers listed in Appendix Table S2. After probe purification with in-house-constructed Sephadex G-50 columns, the probe was added to the tubes containing the membranes and hybridized overnight at 65°C. Non-specifically bound probe was removed by washing the membranes twice in 1× SSC/0.1% SDS and once in 0.2× SSC/0.1% SDS, 0.1× SSC/0.1% SDS and 0.05× SSC/0.1% SDS for 20 min each at 65°C. Membranes were typically exposed to X-ray film for 1 h–1 week. For quantification, images were acquired using a FLA 7000 phosphorimager (GE Healthcare). Levels of the 18S and 25S rRNA species measured by ethidium bromide staining were used as loading controls. All Northern blotting experiments were repeated ≥ 2 times with independent biological samples.

## RNA fluorescence *in situ* hybridization (RNA-FISH)

Fifty millilitres yeast culture was grown in YPD to > 0.45 $OD_{600}$ before transfer to YPG for 2 h. 50 ml of cells at $OD_{600}$ 0.6 was pelleted (900 $g$, 4 min) and fixed with 4% (v/v) paraformaldehyde in PBS (45 min, 80 rpm, 22°C). Fixed cells were washed twice with 10 ml FISH buffer A [1.2 M sorbitol, 0.1 M $KHPO_4$ (pH 7.5)] and re-suspended in 1 ml FISH buffer B (FISH buffer A, 20 mM ribonucleoside vanadyl complex (VRC), 20 µM 2-mercaptoethanol). The mixture was incubated (15–40 min) at 30°C with 15 µl lyticase (25 U µl$^{-1}$, Sigma) until > 70% of cells were spheroplasted, as observed by microscopy. Cells were pelleted (900 $g$, 3 min, 4°C) and washed with and then re-suspended in 1 ml FISH buffer B without 2-mercaptoethanol. ~150 µl of cells was left to settle (30 min, 4°C) on poly-L-lysine treated coverslips. These were gently washed with 2 ml FISH buffer A to remove unattached cells and incubated (−20°C, > 3 h) in 2 ml 70% ethanol. Samples were rehydrated twice with 2 ml of 2× SSC for 5 min at room temperature and washed with 40% formamide in 2× SSC. For the hybridization, 0.5 ng of each probe, 10 µg of *E. coli* tRNA and 10 µg of salmon sperm DNA were mixed and lyophilized in a SpeedVac. 12 µl of 40% formamide, 2× SSC, $NaHPO_4$ pH 7.5 was added, and the probes were denatured at 95°C for 3 min followed by the addition of 12 µl of 2× SSC, 2 mg ml$^{-1}$ BSA, 10 mM VRC. Hybridization was performed overnight at 37°C in a parafilm-sealed chamber, where the coverslips with the cells facing down were placed onto 22 µl of the hybridization mixture. The coverslips were then subjected to a series of washes: twice with 40% formamide/2× SSC (15 min, 37°C); once with 2× SSC, 0.1% Triton X-100 (15 min, 22°C); once with 1× SSC (15 min, 22°C); and once with 0.05× SSC (15 min, 22°C). The Stellaris GFP probes were incubated overnight at 30°C, then washed several times in a 10% formamide solution and stained with a PBS solution containing DAPI. The coverslips were dipped

into $H_2O$. Once dry, coverslips were mounted onto a microscope slide using ProLong Diamond Antifade Mountant with DAPI (Life Technologies), allowed to polymerize for 24 h in the dark and then sealed with nail varnish. Cells were imaged using a DeltaVision CORE wide-field fluorescence deconvolution microscope using a 100×/1.40 objective lens. 21–31 0.2 µm z stacks were imaged with an exposure time of 0.01 and 1 s for DAPI and Cy3 channels, respectively. All RNA-FISH experiments were repeated ≥ 2 times with independent biological samples.

## RNA-FISH probe design and synthesis

For *GAL1*, DNA probes of ~50 nt and ~50% GC content were designed with five modified bases (amino-allyl dT) spaced by about ~10 nt included for the incorporation of the fluorophore (see Appendix Table S3 for probe sequences). Modified DNA oligos were custom ordered from MWG Eurofins. For the labelling of the probes, a total of 5 µg was purified using the QIAquick Nucleotide Removal Kit (Qiagen) and eluted with 40 µl of $H_2O$. The probes were then lyophilized in a SpeedVac, re-suspended in 10 µl of 0.1 M sodium bicarbonate pH 9.0 and added to the dye-containing tube (CyDye™ GE Healthcare, Cy3 PA23001). The tube was vortexed vigorously followed by a quick spin. The reaction was incubated overnight at room temperature with low speed shaking. The probes were purified using the QIAquick Nucleotide Removal Kit (Qiagen) and eluted with 100 µl of elution buffer (supplied with the kit). The concentration and efficiency of the labelling was measured using a spectrophotometer. Probes were stored in the dark at −20°C. The labelling efficiency was calculated as described (Zenklusen & Singer, 2010). The remaining genes were fused at their 3′ ends to S65T pFA6a-GFP sequence (Longtine *et al*, 1998), and the common GFP sequence detected using 27 × 20 nt probes labelled with Cy3 fluorophore obtained from Stellaris. The GFP probes were washed in a PBS/DAPI solution and were mounted in ProLong Gold without DAPI.

## Yeast strains

All *S. cerevisiae* strains used in this study are listed in the Appendix Table S6. All strains and genetic manipulations were verified by sequencing or PCR-based methods.

## Mathematical modelling

RNA synthesis and degradation were modelled as described in the main text. Four of the five parameters were sampled via Latin Hypercube with 1,000,000 sampling points. The degradation rate was sampled as described below. 10,000 cells were simulated for 500 min to reach steady state and the number of nuclear and cytoplasmic RNA recorded. The Kolmogorov–Smirnov statistic was used as a goodness-of-fit metric to compare simulated results to raw data. The best 10,000 parameter sets as judged by fit to nuclear and cytoplasmic RNA were then taken forward. For each strain, the modal value of the histogram of the mean initiation rate (on * init/ (on + off)) was taken from the fits to the cytoplasmic data. The values for the nuclear processing rate were then determined by sampling data points from the ratio of mean initiation rate to nuclear export rate giving the fits to the nuclear data. Again, 10,000

parameters were sampled from the values determining the nuclear distributions, following the mean initiation rate distribution given by the cytoplasmic data.

## Quantification and statistical analysis

### Bioinformatic analysis

**Identification of sense and antisense TSSs in yeast and humans** Cap analysis of gene expression (CAGE) data in HeLa cells was obtained from the ENCODE repository on the UCSC Genome Browser (Rosenbloom *et al*, 2010), and used to determine genome coordinates of sense and antisense transcript start sites (sTSSs and asTSSs, respectively). Data were pooled from nuclear and cytoplasmic fractions, both polyadenylated and non-polyadenylated, and from whole cell extract, for which only polyadenylated data were available. CAGE cluster coordinates, determined with an HMM algorithm applied to the CAGE tag data, were obtained from the same source. To determine TSS coordinates in HeLa, we took the same approach presented previously by Conley and Jordan (2012). Clusters were ignored if they contained less than two overlapping CAGE tags, as it has been previously reported that two or more overlapping tags represent validated TSSs (Carninci, 2006; Faulkner *et al*, 2009). The TSS coordinate of a given cluster was taken to be the base with the highest density of mapped CAGE 5′ ends. As an added step, TSSs were excluded from all subsequent analyses if they contained less than three NET-seq reads in a 200-bp window immediately downstream, within the same orientation. To determine the sTSS of a given protein-coding gene, we scanned within a region 500 bp upstream of the left-most annotated TSS, and 500 bp downstream of the right-most annotated TSS. In the case of multiple sTSSs, the one with the highest CAGE density was taken to be the predominant sTSS. asTSSs were determined by scanning between the sTSS and the annotated transcript end site; again, the asTSS with the highest CAGE density was considered the predominant asTSS in the case of multiple candidates.

Budding yeast sTSSs and asTSSs were determined using transcript isoform sequencing (TIF-seq (Pelechano *et al*, 2013), using the list of major transcripts provided, supplemented with the list of cryptic transcripts from Neil *et al* (2009). For each gene, the sTSS was derived from the sense transcript which had the highest number of supporting NET-seq reads in YPD, and which completely encompassed the open reading frame. The asTSS was taken as the antisense transcript with the highest number of supporting NET-seq reads, and which overlapped the open reading frame in the antisense orientation.

To assess whether asTSSs were better aligned to the sTSS or the end of the 1st exon in HeLa, we determine for each gene the distance between the sTSS to the asTSS, and expressed it as a fraction of the distance between the sTSS and the end of the 1st exon. We compared the resultant histogram to a randomly generated distribution, in which the asTSS for each gene was randomly reassigned to a base within the region shown in Fig 1D. This approach was repeated in HeLa using the end of the 2nd exon in place of the 1st, to assess whether asTSS showed preferential alignment to the 2nd exon over the sTSS. It was also repeated in yeast, using the 3′ end of the open reading frame in place of the end of the 1st exon (Fig 1G).

**Correlating sense and antisense transcription** NET-seq data were obtained in HeLa cells from Nojima *et al* (2015), specifically their data obtained using an antibody against all forms of Pol II, phosphorylated and unphosphorylated. NET-seq data in yeast were obtained from Churchman and Weissman (2011). To compare sense transcription levels between genes with and without an asTSS, we calculated the average number of NET-seq reads per base pair within the 1st exon, and compared the distribution between the two gene groups using a Wilcoxon rank sum test. Correlations between the transcription levels of different sorts of transcript were calculated by determining the Spearman correlation coefficient between the numbers of NET-seq reads in the 300-bp windows shown in Fig 2C and D. The same approach was taken with the GRO-seq (for HeLa) and PRO-seq data (for yeast), obtained from Core *et al* (2014) and Booth *et al* (2016), respectively.

**Assessing ChIP levels around sense and antisense TSSs** Genome-wide levels of histone modifications and nucleosome occupancy were obtained from the following sources: For budding yeast, genome-wide levels of H3K36me3 and H3K79me3 were from Kirmizis *et al* (2009) (GSE14453). Levels of H3K4me1 and H3K4me3 were from Kirmizis *et al* (2007) (GSE8626). Levels of H3K9ac and H3K27ac were from Weiner *et al* (2015) (GSE61888). Nucleosome occupancy levels were from Kaplan *et al* (2009) (GSE13622). Genome-wide levels of gene compaction, determined using Micro-C, were from Hsieh *et al* (2015). Levels of H3K9ac in deletion strains of various histone modifying enzymes were from Weinberger *et al* (2012) (SRA051855.1). For HeLa cells, genome-wide levels of H3K36me3, H3K79me3, H3K4me1, H3K4me3, H3K9ac and H3K27ac were obtained from the ENCODE experiment matrix. Nucleosome occupancy levels were from Kfir *et al* (2015) (GSE65644). Levels of H3 histone modifications are not normalized to levels of H3. We assessed average levels only in genes with an asTSS, comparing the two quintiles with the highest and lowest levels of antisense transcription (determined by NET-seq) in a 300-bp window placed immediately downstream of the sTSS. We wished to simultaneously assess levels upstream of the sTSS, downstream of the asTSS and in the region between both TSSs. To account for the varying distances between sTSS and asTSS, we broke this region into a hundred bins, calculating the average ChIP level within each bin.

**Comparing gene compaction levels** Gene compaction levels in budding yeast were obtained from Hsieh *et al* (2015). Different gene groups were compared as discussed in the Results.

### RNA-FISH analysis

**Software** Image quantification was performed using custom Matlab (MATLAB Statistics and Image Toolboxes Release 2015a, The MathWorks, Inc., Natick, MA, USA) scripts based in part on elements of FISH-quant (Mueller *et al*, 2013) and CellProfiler (Carpenter *et al*, 2006), and utilizing MIJI (https://imagej.net/Miji) and MIJ (http://bigwww.epfl.ch/sage/soft/mij/) to import data from FIJI (Schindelin *et al*, 2012). The custom scripts allowed for greater automization of the quantification process than is possible with FISH-quant and the algorithms were tailored to our data.

Due to slight differences in experimental protocol, including model of microscope, between the data sets including the engineered forms of *GAL1* and the sets including GFP-tagged genes parameters and methods for analysis also differed between these groups. In the following description, the engineered constructs expressing high or low antisense at *GAL1* and with or without the *set3Δ* mutation will be referred to as the *GAL1* set and the GFP-tagged genes (*TIM17, ATP4, GCV3, URA4, HMS2*) will be referred to as the endogenous set.

**Deconvolution and background subtraction** Images were deconvolved with a conservative deconvolution method and 10 cycles (*GAL1* set) or 15 cycles (endogenous set) using DeltaVision Softworx software. DAPI and Cy3 channels for the images were processed separately. Images were background-corrected with the following procedure. The median of all pixel intensities for each channel and each image, $p_{med}$, was found. This was chosen as it was observed to generally be close to the modal value of the distribution. A measure of the spread around this value was also found by constructing a metric similar to the standard deviation from all pixels with intensities less than or equal to the median intensity. The median plus this spread value was taken to be the background value and subtracted from all pixels, background = $p_{med}$ + sqrt[$(1/(N_i–1))$ $\sum_i((p_i – p_{med})^2)$], where $i$ runs over pixels with intensity less than or equal to $p_{med}$, $N_i$ is the number of pixels with intensity less than or equal to $p_{med}$, and $p_i$ is the intensity of pixel $i$. Thus, the new intensity of $p_j$ was $p_j$—background where $j$ runs over all pixels. Any pixels that had negative intensity following this were set to 0.

**Foci identification and thresholding** Candidate foci in the FISH (Cy3) channel were initially identified using Piotr's Matlab toolbox (https://pdollar.github.io/toolbox) nonMaxSupr function with a 1 pixel radius for detection. Images from each biological repeat and strain were processed together. To distinguish foci from random clustering of fluorescently labelled molecules, foci intensities were compared between the strains under testing and a double knockout strain, *gal10-1ΔΔ*, which has no sequences to which the FISH probes should hybridize. In each experiment, histograms of all foci intensities as returned by the nonMaxSupr algorithm were constructed (bin width 250 for the *GAL1* set and 50 for the endogenous set, normalized by probability). For each strain, a tentative intensity cut-off was taken to be the first bin in which there was 10 times more signal in the strain than the knockout (with manual adjustment for obvious outliers). Within each experiment (which could contain multiple strains), the final cut-off value was taken as the mean of the tentative cut-off values. All foci with intensities less than the cut-off value from a set were not considered in all further analysis.

The deconvolution, background subtraction and foci identification are performed by the supplied FindAndAnalyseFoci Matlab function. This function should be run on all strains and the knockout from a single experiment before a threshold for valid foci is determined. This threshold can then be found using the DetermineCutoffs Matlab script.

**Automated nuclei detection** A separate script automatically identifies nuclei and cells and quantifies the foci that fall within the nuclear and cellular boundaries. The first step of the process is to identify the nuclei in three dimensions using the DAPI channel. The procedures followed here allowed for improved detection of individual nuclei that differed in brightness or were very close to neighbouring nuclei. The DAPI channel of each image was scaled to the minimum and maximum intensity pixels, that is $p_i = (p_i – \min(p))/(\max(p) – \min(p))$, where $i$ runs over all pixels, $\min(p)$ and $\max(p)$ denote the minimum and maximum pixel intensities of the set, respectively. The scaled DAPI images then have a Gaussian filter applied using Matlab function imgaussfilt3 with a smoothing-kernel standard-deviation value of 2 for the *GAL1* set and 2.5 for the endogenous set. At this point, the analysis for the two sets diverged considerably and so they are described separately. For the *GAL1* set, for each processed image, Otsu's method for multiple thresholds, Matlab function multithresh, was used to give six threshold levels and the image was segmented into seven levels around these using the Matlab function imquantize. The segmented images thus contained pixels with values from 1 to 7. Each segmented image was then restricted to a subsection of the available $z$ stacks by setting any pixels in $z$ planes below or above certain values to zero, to avoid any errors due to using overly blurred portions of the image. For images with 31 $z$ stacks, images were typically restricted to include only pixels from $z$ planes 12 to 22, inclusive, and for images with 21 $z$ stacks, images were typically restricted to $z$ planes from 2 to 20, inclusive. These values were manually adjusted in some cases to allow for off centred focusing, but the same planes were used for all images of a particular strain taken in a single experiment.

The segmented levels were cycled through from the 3rd to the 7th levels. For each level, a 3D logical image was formed from the pixels with value equal to the value of the level. The $z$ stacks were then cycled through and all holes (areas with pixels with value 0 inside areas with value 1) were filled, Matlab function imfill with flag "holes". Then, all 3D-contiguous regions (with 26 connectivity) of pixel value 1 that had more than 6,000 or fewer than 50 pixels were removed by setting all pixels in the region to zero, using Matlab xor and bwareaopen functions. Any remaining 3D-contiguous regions (26 connectivity) were then labelled using the Matlab function bwlabeln. For levels lower than the final level, an identical procedure was performed on the level immediately above, without the final labelling step. Each labelled section was then cycled through, and if there was no overlap with any non-zero valued pixels in the segmented level above, it was deemed as a good candidate for a nucleus and saved (the centroid was determined with the Matlab function regionprops and flag "centroid", and each coordinate was rounded to the nearest integer). Any overlap with the level above signified the existence of a superior candidate or superior candidates in this region, and this potential nucleus was not saved. This process was repeated until the final level in which no checking against a higher level was possible.

Once all good nuclear candidates had been identified in this way, any nuclei that were very close together were merged with the following procedure. The Euclidean distances, measured in pixel coordinates, between all centroid locations were calculated. A list of all non-equal pairs of centroids that had a distance of less than or equal to six between them was created. This was done by having two nested loops: the outer loop cycled over the centroids from $i = 1$ to $(N_c–1)$ and the inner loop cycled over $j = (i + 1)$ to $N_c$, where $N_c$ is the number of candidate centroids. Any $i, j$ pairs from

this loop with distance less than or equal to six were listed. If any centroid appeared more than once in the list, this list was reordered in ascending order of distances. The ordered list was then cycled through in order and, starting with the *i* element of the pair, if this centroid was repeated, all but this first appearance of this centroid in the list was deleted, and then, the same test and deletion were done with the second centroid of the pair. Then, all pairs of centroids on the list were merged by taking the average of their coordinates and rounding each coordinate to the nearest integer. The distances between the new centroids were calculated, and the process was repeated until no centroids that were within a distance of six from another remained. This procedure prioritized merging centroids that were closest together in the case that there were multiple possible mergers.

For the endogenous set, the initial part of the nuclei detection was much simpler. Each image was divided using two threshold levels, Matlab function multithresh, and segmented into three levels with imquantize. Each segmented image was restricted to only pixels from *z* planes 1 to 20 inclusive. This amounted to either all of the planes or all but the final plane for this set. New logical images were constructed from the restricted pixels with value equal to the highest segment value, 3. The *z* stacks of these images were then cycled through, and holes were filled with imfill (flag "holes" as before). Then, all 3D-contiguous regions (with 26 connectivity) of pixel value 1 that had more than 40,000 or fewer than 600 pixels were removed by setting all pixels in the region to zero, using Matlab xor and bwareaopen functions. The centroid for each labelled region was calculated using the function regionprops with flag "centroid".

At this point, the analysis for the *GAL1* and endogenous sets reconverges in procedure. The list of centroids generated in one of these ways was then used to generate 3D masks of the nuclei with the following procedure. The centroids were cycled through and the mean intensity of the 27 pixels, for the *GAL1* set, or 125 pixels, for the endogenous set, surrounding and including the centre pixel was taken, using the filtered and *z* stack restricted DAPI signal. Logical images were formed for each centroid by setting all pixels with intensity greater than or equal to 0.65 times this mean value to unity and all others to zero. The 26-connected 3D component from this that overlapped with the centroid position was taken as the 3D nuclear mask for this centre point. To avoid counting areas that were too low intensity relative to the image, nuclei that had a mean intensity of the 27 or 125 pixels less than 0.025 were rejected. Once this list of nuclear masks had been created, a further filtering was done by removing any nuclei that had a volume of fewer than 50 pixels for the *GAL1* set or 200 pixels for the endogenous set. For later analysis, additional 2D nuclei masks were formed by taking the maximum of the 3D nuclei through the *z* stacks which due to the masks being stored as logical images corresponds to the greatest extent in the *x* and *y* coordinates that the nucleus has in any of the allowed *z* stacks. At this step, the 2D nuclei were relabelled based on connected components with eight connectivity (Matlab function bwlabel) as it is possible for 3D nuclei to not touch but to overlap when flattened in this way. For later classification of foci, these 2D nuclear centres were extruded to fill the allowed *z* stacks.

**Automated cell detection** The 2D nuclei identified above were used as seed points to identify cell outlines. Cell masks were identified using the CellProfiler function IdentifySecPropagateSubFunction

from the MEX compiled file supplied with the developer's version of CellProfiler 1.0. Cells were identified with a combination of the DAPI signal, FISH foci signal and autofluorescence of the cells observed in the FISH channel. The IdentifySecPropagateSubFunction takes a number of inputs: a set of seed points for cells; a 2D image with varying intensity; a 2D logical image; and a regularization factor which determines how to weight between the 2D images and distance to the nearest seed points when determining cell boundaries. A regularization factor value of 0.0001 was used in all cases.

The 2D images with varying intensities were constructed by combining processed DAPI and FISH channels in the following way. For each channel of each image, the sum in the *z* direction was taken to get a flattened image and the maximum and minimum intensities observed in this image were found. Each pixel in the image was then normalized as $p_i = (p_i - \min(p))/(\max(p) - \min(p))$. For each image, the normalized flattened channels were averaged, and then, this average had a Gaussian filter applied (Matlab function imgaussfilt with smoothing-kernel standard-deviation value of 2).

The 2D logical images were constructed starting in a similar way by creating normalized and flattened images as above prior to the averaging step. Each channel for each image had a Gaussian filter applied; the DAPI signal used a smoothing-kernel standard-deviation value of 5 and the FISH channel used a value of 2. Each filtered channel for each image then had a threshold generated by taking the lowest value from a three-level Otsu's method thresholding (Matlab function multithresh with three levels). Each of the filtered channels for each image was then converted into a logical image with pixels having an intensity less than the threshold being set to 0 and the rest being set to 1. The channels for each image were then combined using the logical OR operation.

The nuclei and cells were then further processed to remove anything touching a border of the image as cells touching the border are likely to have part of the cell out of the image and using them for data collection could bias the results. First any 3D-resolved nuclei that touched the border in 3D were removed (Matlab function imclearborder). Following this, any cell masks that were touching the border were removed. Then, any 3D nuclei considered as being too large (i.e. containing more than 3,000 pixels for the *GAL1* set or 20,000 pixels for the endogenous set) were removed. The size-based removal was done after cell detection as large detected nuclei often corresponded to multiple nuclei close together and keeping these causes the corresponding cells to be detected as one large cell, which often aided in assigning the correct boundaries to nearby cells. Then, any cells that did not overlap with any nuclei were cleared and any nuclei that did not overlap with a cell were also cleared. Finally, any cells that had two or more nuclei were removed along with the corresponding nuclei. This case can happen rarely when 3D nuclei do not touch but overlap when collapsed onto 2D resulting in a merged nucleus. These were removed as it is likely that there would be some cytoplasmic overlap also in this case.

**Foci classification and mRNA quantification** Cells were first divided into nuclear and cytoplasmic components. The nuclear components were formed first by flattening the 3D-resolved nuclei masks (taking the maximum over the *z* stacks) and then extruding them to the allowed *z* stacks (the same *z* stack limits used when restricting in the nuclei detection phase). The 2D cell masks were

then extruded to the same allowed *z* stacks and had the nuclear area within them set to zero, which gave the cytoplasmic components. Foci whose centre pixel lay in the nuclear region where classified as nuclear transcripts and foci whose centre pixel lay in the cytoplasmic region were classified as cytoplasmic transcripts. Any foci lying outside these regions were discounted from all further analysis.

In order to quantify the foci in terms of RNA molecules, an intensity value for all accepted foci was calculated by taking the mean of the 27 pixels immediately surrounding and including the central pixel (found by rounding each coordinate of the output of the nonMaxSupr function to the nearest integer). For a strain and image set from a single experiment, the median of these intensity values was calculated and was taken to correspond to a single RNA molecule. There is no reason that a FISH focus should contain only a single RNA, especially in the nucleus, but we assumed that foci most commonly contained a single RNA. Each focus intensity was then converted into a corresponding number of mRNA molecules by dividing by the median intensity and rounding to the nearest integer. Note that this can result in foci being classified as containing zero RNA molecules and provides an additional filtering step similar to the initial cut-off based on knockout strains.

The detection of nuclei and cells, and the quantification of foci is performed by the supplied Matlab script DetectCellsAndQuantifyFoci. This script should be run after the FindAndAnalyseFoci function and DetermineCutoffs script as it uses the data generated in the first script and the cut-off generated after averaging the determined cut-offs over an experiment. The cut-off must be manually changed in the DetectCellsAndQuantifiFoci script before running. Scripts and images are available from https://doi.org/10.17632/dhnvj4xs5d.1.

*Chromatin immunoprecipitation*
Real-time quantitative PCR (qPCR) was performed using a Corbett Rotorgene and SYBR green mix (Bioline). qPCR was performed in triplicate for each sample and quantified using a standard curve. Histone modification data [(IP—no antibody control)/input] were expressed as a percentage of the input and normalized to levels of histone H3 at each amplified region. Data are presented as averages of ≥ 2 biologically independent experiments, with error bars representing the standard error of the mean.

*Northern blotting*
Raw images for quantification were captured using a FLA 7000 phosphorimager (GE Healthcare). Mean intensity of band was quantified using Fiji/ImageJ. Normalized levels of RNA were obtained from Northern blots. Data were tested for normality using a Kolmogorov–Smirnov test; all *P* values were > 0.7 indicating no evidence to suggest the data were not normally distributed. As such, unpaired *t*-tests were used to compare levels of RNA between strains across multiple experiments. Degradation rates were obtained by fitting results across six timepoints to an exponential using MATLAB fit function. Root mean square error was taken as goodness-of-fit metric and correspondingly used as the standard deviation of the estimator of the exponential decay term. To sample degradation rates for the purposes of modelling, degradation rates were sampled from a Beta distribution with maximum and minimum values given by the 95% confidence intervals of the estimator

with standard deviation equal to the root mean square error of the estimator.

**Data availability**

Images for RNA-FISH experiments, computer codes and all source data are available from Mendeley https://doi.org/10.17632/dhnvj4xs5d.1. Computer code is provided as Computer Code EV1.

**Expanded View** for this article is available online.

## Acknowledgements

We thank the J.M. and A.A. laboratories for critical discussions, Anitha Nair for excellent technical support, Simon Haenni for 5′ and 3′ RACE mapping of the transcripts, and Ilan Davis and Micron Oxford for microscopy support. This work was supported by: The Wellcome Trust (WT089156MA to J.M.); the BBSRC (BB/P00296X/1 to J.M.); the Leverhulme Trust (RPG-2016-405 to J.M.); a Wellcome Trust Strategic Award (091911) supporting advanced microscopy at Micron Oxford (http://micronoxford.com); EPSRC and BBSRC studentships (EP/F500394/1 to T.B.; EP/G03706X/1 to S.R.; BB/J014427/1 to E.S.; BB/M011224/1 to P.L.) and a Royal Society University Research Fellowship (UF120327 to A.A.).

## Author contributions

Project conception: JM and AA; strain construction, *GAL1* RNA-FISH, chromatin immunoprecipitation, Northern blots: FSH; other RNA-FISH: MW; GFP tagging of strains: PL; bioinformatics: SCM; modelling: TB, SR, ES and AA; image analysis: AA. All authors involved in interpretation of the data. JM and SM wrote the paper with input from FSH, TB and AA.

## Conflict of interest

JM acts as an advisor to and holds stock in Oxford Biodynamics plc., Chronos Therapeutics Ltd., and Sibelius Natural Products Ltd.

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
