## [Review Process File · Molecular Systems Biology]

Antisense transcription-dependent chromatin signature modulates sense transcript dynamics

Thomas Brown, Françoise S. Howe, Struan C. Murray, Meredith Wouters, Philipp Lorenz, Emily Seward, Scott Rata, Andrew Angel and Jane Mellor

Review timeline:

Submission date:	26 September 2017
Editorial Decision:	24 October 2017
Revision received:	17 December 2017
Editorial Decision:	9 January 2018
Revision received:	13 January 2018
Accepted:	16 January 2018

Editor: Maria Polychronidou

Transaction Report:

1st Editorial Decision

24 October 2017

Thank you again for submitting your work to Molecular Systems Biology. We have now heard back from the three referees who agreed to evaluate your study. As you will see below, the reviewers acknowledge that the main findings seem interesting. They raise however a series of concerns, which we would ask you to address in a revision of the manuscript.

Without repeating all the points listed below, one of the more fundamental issues was raised by reviewer #2, who recommends further analyses in order to support more convincingly the conclusions derived from the analysis of the GAL1 locus.

I wanted to clarify that follow-up analyses of the mechanisms underlying the effect of antisense transcription on sense transcription are not mandatory for the acceptance of the study.

All other issues raised by the reviewers would need to be convincingly addressed.

REVIEWER REPORTS

Reviewer #1:

Brown et al. use an impressive complement of experimental approaches to characterize the effect of antisense long non-coding RNA transcription on mRNA expression in yeast and human. The authors use their world-leading excellence in cutting-edge genomics approaches to select top quality data to make their initial discovery of a conserved arrangement of sense and antisense lncRNA start sites in

yeast and human genes. The authors also use this analysis to segment cohorts of genes based on antisense lncRNA transcription level for further analysis. A computational analysis allows the authors to identify a set of conserved chromatin signatures associated with antisense lncRNA transcription in yeast and human. The analyses are of high quality and incorporate many recent data (HiC, NETseq, GROseq, PROseq). The authors derive a stochastic model of transcription to probe the effect of identified antisense lncRNA expression on mRNA expression. From here on, the authors focus on budding yeast as a model. The authors make the exciting discovery of compensating changes of antisense lncRNA expression on mRNA production and stability. Importantly, the model is well-parameterized, and key predictions derived from the mathematical model are experimentally validated using probing genetic engineering of the yeast GAL1 example locus. The authors use RNA Fish which is well suited to address their experimental question. The authors conclude with an impactful mechanistic hypothesis for antisense lncRNA expression: it buffers chromatin structure from modulating effects of the Set3-HDAC complex that has previously been implicated in timing the expression of mRNA expression by lncRNA mechanisms. The authors find support for this important conclusion by characterizing H3K9ac in yeast. All in all, this study offers a high-quality insight into the effect of antisense lncRNA transcription on the corresponding mRNA through a unique combination of powerful experimental approaches. The general conclusions are well supported and the reasoning explained clearly in concise language. The compensating effects of antisense lncRNA transcription on mRNA production and decay represent an important finding, especially because the authors work across experimental systems. The findings offer a helpful consensus conceptual advance to reconcile previously described positive and negative correlations of sense and antisense transcription. This paper will hence be relevant for a wide readership in several experimental systems, which appears to be an excellent fit with MSB. The manuscript is based on a large volume of high-quality analysis to arrive at an insightful conclusion. I believe the points are well supported and so only minor revisions seem necessary before publication.

Major points:

N/A

Minor points:

- It seems like the authors are provided with an opportunity to enhance the clarity of the conclusions that are meant to be derived from the model/graphical abstract. The 4 "cells" appear almost identical and the key concept does not jump out.
- The experimental validation of their modeling and meta-genomic analysis is very valuable. However, as the authors understandably restrict this analysis to a single locus in yeast. It may be advisable to word some paragraphs a little more carefully as the authors are not in the position to be particularly confident that these findings translate to human cells, where such an analysis has not been performed as it would be beyond the scope of this article.
- The model in which the Set3 HDAC modulates transcription dynamics predominantly at low antisense genes is intriguing. Perhaps the authors are in the position to test if dynamics at high-antisense genes could be mediated by the Rpd3s HDAC complex that may be recruited co-transcriptionally through antisense lncRNA transcription? Relevant histone acetylation data has for example been described in Venkatesh et al., Nature 2012. NETseq data in *rcol* mutants has been described in Churchman et al., Nature 2011.
- The authors focus their manuscript on the effect of antisense lncRNA transcription. However, the chromatin signatures at gene TSS may also be influenced by divergent lncRNA transcription as indicated in Fig. 1A. At least in budding yeast it is clear that divergent lncRNA transcription is connected to chromatin regulation (Marquardt et al., Cell 2014), perhaps this aspect could be added in the discussion.
- Gene regulation by antisense lncRNA transcription has been observed beyond human and yeast. Perhaps the authors can add a few sentences for which other species with evidence for antisense lncRNA regulation their findings may apply (*Neurospora*, *S. pombe*, plants ...)? The increased mRNA stability could be discussed in relation to the chance for dsRNA formation that may trigger siRNA formation and could conceivably have destabilizing effect in organisms with active small RNA machinery (e.g. not budding yeast).

Reviewer #2:

Summary

The paper by Brown et al. aims at describing the links between antisense-dependent chromatin structure and sense transcription dynamics. In the first part of the publication, the authors establish that antisense transcription leads to a specific chromatin structure including nucleosome positioning and histone modifications. Importantly, the antisense-dependent chromatin structure of genes is similar in *S. cerevisiae* and HeLa cells suggesting a nice conservation of these features. Given that they observe no correlation between sense and antisense nascent transcription, they use data obtained from the analysis of an engineered GAL1 gene combined with mathematical modeling to propose that antisense transcription decreases the rate of sense production, processing and degradation. At last, they show that a Set3 histone deacetylase mutant can modulate these parameters, even in the absence of antisense transcription. From this, they conclude that antisense transcription modulates sense transcription through its impact on chromatin structure. However, the underlying mechanism is pretty obscure.

This publication deepens earlier observations from this lab (Murray et al., 2015), mainly using bioinformatics analyses of published data but also through a series of smFISH and half-life measurements of the GAL1 transcripts produced from engineered GAL1 genes exhibiting either high or low AS RNA levels as described earlier (Murray et al., 2015). A huge part is also devoted to the mathematical modeling of transcription dynamics using GAL1 as a model gene.

General remarks

The first part, concerning the importance of antisense transcription in chromatin structure is quite convincing. These data are important for the non-coding RNA community and of particular interest for the chromatin structure field. However, some of the *S. cerevisiae* observations, although less complete, were already present in Murray et al., 2015. On the other hand, extension of these analyses to HeLa cells is novel. The mathematical modeling is less convincing for reasons explained below. At last, the part on the Δ set3 mutant recapitulating chromatin features of genes with high AS is difficult to read and the schematic outlining at the end gives no hint about the potential mechanism relating the effects of Set3 and AS transcription on mRNA production and stability, or how AS transcription may buffer chromatin against the modulating effects of Set3 during sense transcription. Overall, while some observations are interesting, there is some redundancy with already published data (Murray et al., 2015) and the mathematical model based on the analysis of a single engineered locus needs further confirmation on natural genes.

Major points

- One critical point is the use of the TIF-seq data from Pelechano et al., 2014 for the definition of antisense-containing genes. Since these data were collected in WT condition, only the most stable antisense transcripts are identified and many highly unstable antisense transcripts, which can be detected by nascent transcription analysis, are not included in this list. Do results stay consistent if antisense containing-genes are selected through NET-seq (or Pro-seq) criteria? For example, Figure 5A shows that genes with antisense produce more stable sense transcripts than genes without antisense. Is this still true if antisense-containing genes are considered based on nascent transcription and not TIF-seq?
- Figure 2A: It would be interesting to show the nascent transcription profiles considering the high/low levels of sense transcription and high/low levels of antisense transcription. This information is important to fully appreciate the nucleosome and histone modification profiles shown in Figures 2B and 3.
- Figure 4 and S3: by mathematical modeling, the authors propose that GAL1 gene transcription is decreased in presence of high AS. This prediction should be confirmed by a direct experiment.
- The authors suggest that their observations may reflect a general mechanism. However there is a discrepancy with data already published by Eser et al., 2016. Using 4-ThioU, this paper proposes that in *S. pombe* antisense transcription slows down mRNA synthesis but does not affect mRNA stability. For these reasons, besides the data from Wang et al. 2002 used in Figure 5A, it may worth also analysing the data from Sun et al. 2013 to confirm that genes with AS transcription produce mRNAs with a longer half-life.
- The GAL1 locus may not be the optimal choice as a model gene since it has a complex profile of ncRNA transcription. Moreover, the upstream GAL10 ncRNA (ncRNA upstream of GAL1) may be more important than the GAL1 antisense since its repression leads to increase of GAL1 expression (Houseley et al., 2008) (which is not the case of GAL1 antisense repression). Thus, GAL1 presents a

complex ncRNA pattern and is not solely influenced by its antisense. The authors should confirm their model by experimental analysis of natural genes presenting either high or low levels of AS transcription.

Minor points

- Figure 2B : not clear whether the genes with high/low sense are taken among the total number of genes, or among the 1500 AS producing genes ? Do the high/low sense genes really produce no AS ?
- Figure 2E : Why analyse 1045 and 1046 genes with low versus high S, and not the same 300 as in the previous figure ? Which genes were chosen in this case ? They have no AS ?
- p.6 : « there is no difference in gene activity », is misleading since it may be understood as transcriptional activity, which is proposed later to be decreased in presence of high AS.
- In the previous publication (Murray et al., 2015), most of the metagene profiles were normalized to H3, which is not the case here (Figures 3 and S2). Why this difference? In contrast, the ChIP results are normalized to H3 (Figure 6 and S5); in this case the H3 levels should be shown since based on earlier data, they are not the same in the 2 strains expressing high and low GAL1 AS (Murray et al., 2015).
- Figure 4 : how can one exclude that GAL1 mRNA half-life is not affected by the terminator and the mutations therein in an AS independent way. Where does sense transcription exactly end? How is sense 3' cleavage and polyadenylation affected by the mutations in the terminator?
- Figure 4D : At what time-point was sense RNA measured. Why normalize to high AS and not some other internal control?
- p. 10 : "H3K9Ac may result from but is not causally related to histone turnover". This statement sounds contradictory. If an event results from another event, these events are causally related.
- Fig. 6D : the levels of H3K9Ac are higher at the GAL1 gene with high AS and the sense transcription rate is lower compared to GAL1 with low AS (Figures 4 and 5). Does the lower GAL1 sense transcription rate (in GAL1 high AS) contribute to the higher H3K9Ac levels by recruiting less histone deacetylases?

Reviewer #3:

In this manuscript Brown et al. start by describing the chromatin signature associated to antisense (AS) transcription in budding yeast and humans. Interestingly they find an association between antisense expression and decreased chromatin compaction. In parallel to this, the authors investigate the effect of AS transcription by modeling transcription of a sense gene and fitting their model with data obtained from single-molecule RNA-FISH. In particular they use a system that they have previously developed where the same gene (GAL1) can be expressed with either high or low level of AS transcript. Combining this data with measures of mRNA stability, they identify that although the abundance of mRNA does not change significantly, the general turnover of GAL1 is accelerated in the absence of antisense transcription. Following this initial observation, the authors hypothesize that some of the observed chromatin marks identified in their first part could be involved in this process. By deleting SET3 the authors are able to mimic in the strain with low AS a decreased RNA turnover (similar to the strain with high AS). Using this information, the authors conclude that antisense transcription, by modifying the chromatin status, is able to affect the posttranscriptional life of the mRNA.

Many researchers are trying to dissect the process by which changes in the nucleus can affect the post-transcriptional life of the RNA. However, we are still far from a clear understanding of this process. I think this work significantly contributes to our understanding of this process. The study of the crosstalk between nucleus and cytoplasm is very challenging due to the high interdependence between all the factors. In this manuscript the authors perform a thoughtful analysis of the GAL1 system. It would be great to expand this study to different loci, or even genome-wide. However, due to the amount of work required for that study, I think it will be out of the scope of the current manuscript. I think this paper is well written and provides insightful information to understand the control of gene expression. However, to improve its clarity and maximize its impact, I have the following questions.

1. The usage of *set3Δ* mimicking the presence of an antisense is very interesting. However, as we

know that *set3Δ* can also generate novel cryptic transcripts (Kim et al 2012 PMID 22959268). Does the theoretically low antisense strain produce high antisense after SET3 deletion? Or is it still behaving as a low antisense strain as expected? A Northern blot or qPCR analysis would clarify this point.

2. In Figures 2 and 3 the authors split the pairs sense antisense according to expression levels of sense or antisense transcripts. In some cases it seems that genes with high levels of antisense are very similar to the genes with low level of sense, and vice versa. However, the expression levels of sense and antisense transcripts are not correlated. It would be very useful to have some examples to show the overlap between those different groups (e.g. some Venn diagram depicting the overlap between the 4 groups). Do the shared genes drive the observed chromatin pattern? Or can it be also observed when studying non-overlapping pairs of sense-antisense? Showing some examples for selected panels would be very informative for the overall conclusion.

3. Previous papers have suggested that sense and antisense transcripts can pair in vivo (eg Wery et al 2016 PMID 26805575). How do the authors discard those that could drive the observed changes in stability? Following the same question, do sense and antisense GAL1 molecules coexist in the same cells? Do the authors have tried to image AS GAL1 using their RNA-FISH approach? Some clarification and extended discussion would be helpful.

4. The Fig 7 could be improved by adding some more details or connection lines between the different players. It is difficult to depict a working model integrating the effect of the AS transcript, increase turnover, and SET3.

Some typos:

- Page 3: The first citation of Fig 3B, should be 3A.
- Page 33: extra "(" in the citation of Longtine et al.

Reviewer #1:

Brown et al. use an impressive complement of experimental approaches to characterize the effect of antisense long non-coding RNA transcription on mRNA expression in yeast and human. The authors use their world-leading excellence in cutting-edge genomics approaches to select top quality data to make their initial discovery of a conserved arrangement of sense and antisense lncRNA start sites in yeast and human genes. The authors also use this analysis to segment cohorts of genes based on antisense lncRNA transcription level for further analysis. A computational analysis allows the authors to identify a set of conserved chromatin signatures associated with antisense lncRNA transcription in yeast and human. The analyses are of high quality and incorporate many recent data (HiC, NETseq, GROseq, PROseq). The authors derive a stochastic model of transcription to probe the effect of identified antisense lncRNA expression on mRNA expression. From here on, the authors focus on budding yeast as a model. The authors make the exciting discovery of compensating changes of antisense lncRNA expression on mRNA production and stability. Importantly, the model is well-parameterized, and key predictions derived from the mathematical model are experimentally validated using probing genetic engineering of the yeast GAL1 example locus. The authors use RNA Fish which is well suited to address their experimental question. The authors conclude with an impactful mechanistic hypothesis for antisense lncRNA expression: it buffers chromatin structure from modulating effects of the Set3-HDAC complex that has previously been implicated in timing the expression of mRNA expression by lncRNA mechanisms. The authors find support for this important conclusion by characterizing H3K9ac in yeast. All in all, this study offers a high-quality insight into the effect of antisense lncRNA transcription on the corresponding mRNA through a unique combination of powerful experimental approaches. The general conclusions are well supported and the reasoning explained clearly in concise language. The compensating effects of antisense lncRNA transcription on mRNA production and decay represent an important finding, especially because the authors work across experimental systems. The findings offer a helpful consensus conceptual advance to reconcile previously described positive and negative correlations of sense and antisense transcription. This paper will hence be relevant for a wide readership in several experimental systems, which appears to be an excellent fit with MSB.

The manuscript is based on a large volume of high-quality analysis to arrive at an insightful conclusion. I believe the points are well supported and so only minor revisions seem necessary before publication.

Major points:

N/A

Minor points:

- It seems like the authors are provided with an opportunity to enhance the clarity of the conclusions that are meant to be derived from the model/graphical abstract. The 4 "cells" appear almost identical and the key concept does not jump out.

We thank the reviewer for this comment. We have adjusted the Figure 7 to make the key ideas clearer.

- The experimental validation of their modeling and meta-genomic analysis is very valuable. However, as the authors understandably restrict this analysis to a single locus in yeast. It may be advisable to word some paragraphs a little more carefully as the authors are not in the position to be particularly confident that these findings translate to human cells, where such an analysis has not been performed as it would be beyond the scope of this article.

We have included analysis of endogenous genes with varying levels of natural antisense to extend our model in yeast (shown below and now part of Figure 5 in the revised manuscript).

In addition, we have modified the text regarding human cells.

- The model in which the Set3 HDAC modulates transcription dynamics predominantly at low antisense genes is intriguing. Perhaps the authors are in the position to test if dynamics at high-antisense genes could be mediated by the Rpd3s HDAC complex that may be recruited co-transcriptionally through antisense lncRNA transcription? Relevant histone acetylation data has for example been described in Venkatesh et al., Nature 2012. NETseq data in *rco1* mutants has been described in Churchman et al., Nature 2011.

We thank the reviewer for this interesting suggestion. We have already published analysis of *rco1* mutants in our 2015 paper:

Antisense transcription: **decreased, unchanged, increased**
in mutant relative to wild type

(All H3 modifications normalised to nucleosome occupancy)

These data clearly conform to our hypothesis that increased antisense transcription in the mutant strain is associated with increased nucleosome occupancy and increased histone lysine acetylation (and *vice versa*) although in the case of the *rco1* mutant, these change are found towards the 3' end of the transcription units, as predicted by the data from the Workman group and others.

We have also done as the reviewer requested and analysed the published data for the *rco1* mutant which is shown above, using the same methodology as for the other HDACs analysed in the paper (Fig EV6) (and included this *rco1* mutant data in Fig EV6). In the *rco1* mutant, genes with low antisense show a greater increase in histone lysine acetylation than genes with high antisense. One potentially interesting scenario is that Rco1 and Set3 influence transcription over different regions of genes, supported by the negative genetic interactions between *rco1* and *set3* mutants, with both buffering the effects of antisense transcription. However, it is important to remember that *rco1* causes increased antisense transcription at 870 genes and these may be low antisense genes in the WT hence the relative difference in histone lysine acetylation at the 3' region of genes with low antisense. We confirm that the *rco1* mutant shows a significant increase in antisense transcription reads compared to the WT (shown in a new panel in Figure EV6) suggesting that the chromatin changes are a consequence of more antisense and that the *rco1* mutant may act by a different mechanism to the *set3* mutant.

While the increased acetylation in the *rco1* mutant may result from increased antisense transcription, this is *not* the case for the *set3* mutant. We confirm using the available data from the Buratowski group that at the *genome-wide* levels (i) there is no difference in the levels of transcripts on the antisense strand of genes in the *set3* mutant (Figure EV6) and (ii) that loss of Set3 does not have a general effect on sense or antisense transcript levels, although as pointed out by Buratowski, Set3 does repress specific cryptic internal promoters at the 5' region of some genes which can result in novel transcripts, including in the antisense direction. However, this is not sufficient to be the

cause of the increased acetylation we observe at the 5' region of low antisense genes.

- The authors focus their manuscript on the effect of antisense lncRNA transcription. However, the chromatin signatures at gene TSS may also be influenced by divergent lncRNA transcription as indicated in Fig. 1A. At least in budding yeast it is clear that divergent lncRNA transcription is connected to chromatin regulation (Marquardt et al., Cell 2014), perhaps this aspect could be added in the discussion.

We have done as requested by the reviewer. Certainly, the conclusion from the Marquardt paper that upstream divergent transcripts are associated with chromatin remodelling (SWI/SNF) and acetylation (in their case H3K56ac) is in line with what we have published.

- Gene regulation by antisense lncRNA transcription has been observed beyond human and yeast. Perhaps the authors can add a few sentences for which other species with evidence for antisense lncRNA regulation their findings may apply (Neurospora, S. pombe, plants ...)? The increased mRNA stability could be discussed in relation to the chance for dsRNA formation that may trigger siRNA formation and could conceivably have a destabilizing effect in organisms with active small RNA machinery (e.g. not budding yeast).

We thank the reviewer for these interesting suggestions. We have tried to expand the interpretation to other species as suggested.

Reviewer #2:

Summary

The paper by Brown et al. aims at describing the links between antisense-dependent chromatin structure and sense transcription dynamics. In the first part of the publication, the authors establish that antisense transcription leads to a specific chromatin structure including nucleosome positioning

and histone modifications. Importantly, the antisense-dependent chromatin structure of genes is similar in *S. cerevisiae* and HeLa cells suggesting a nice conservation of these features. Given that they observe no correlation between sense and antisense nascent transcription, they use data obtained from the analysis of an engineered GAL1 gene combined with mathematical modeling to propose that antisense transcription decreases the rate of sense production, processing and degradation. At last, they show that a Set3 histone deacetylase mutant can modulate these parameters, even in the absence of antisense transcription. From this, they conclude that antisense transcription modulates sense transcription through its impact on chromatin structure. However, the underlying mechanism is pretty obscure.

This publication deepens earlier observations from this lab (Murray et al., 2015), mainly using bioinformatics analyses of published data but also through a series of smFISH and half-life measurements of the GAL1 transcripts produced from engineered GAL1 genes exhibiting either high or low AS RNA levels as described earlier (Murray et al., 2015). A huge part is also devoted to the mathematical modeling of transcription dynamics using GAL1 as a model gene.

General remarks

The first part, concerning the importance of antisense transcription in chromatin structure is quite convincing. These data are important for the non-coding RNA community and of particular interest for the chromatin structure field.

We thank the reviewer for these supportive comments and wholeheartedly agree about their relevance to the broader community.

However, some of the *S. cerevisiae* observations, although less complete, were already present in Murray et al., 2015. On the other hand, extension of these analyses to HeLa cells is novel.

We acknowledge this but felt for clarity a direct comparison between the HeLa data and the yeast data would be best. In addition, we have added additional data to the yeast analysis and some of the data sets we have analysed here are of much higher quality than those used in the 2015 paper.

The mathematical modeling is less convincing for reasons explained below.

Respectfully, we think that the modelling is state of the art – rather the reviewer means that the data that has been modelled is less convincing because it is at a single locus.

At last, the part on the Δ set3 mutant recapitulating chromatin features of genes with high AS is difficult to read and the schematic outlining at the end gives no hint about the potential mechanism relating the effects of Set3 and AS transcription on mRNA production and stability, or how AS transcription may buffer chromatin against the modulating effects of Set3 during sense transcription. Overall, while some observations are interesting, there is some redundancy with already published data (Murray et al., 2015) and the mathematical model based on the analysis of a single engineered locus needs further confirmation on natural genes.

We have confirmed the results of the single locus modelling at GAL1 on more genes by obtaining RNA turnover rates and RNA FISH data then modelling rates of transcript production and processing. This data is shown below and is included in the revised manuscript (Figure 5). This data confirms the relationship between high antisense transcription and reduced transcription elongation/export and supports our model derived at engineered GAL1.

Major points

- One critical point is the use of the TIF-seq data from Pelechano et al., 2014 for the definition of antisense-containing genes. Since these data were collected in WT condition, only the most stable antisense transcripts are identified and many highly unstable antisense transcripts, which can be detected by nascent transcription analysis, are not included in this list. Do results stay consistent if antisense-containing-genes are selected through NET-seq (or Pro-seq) criteria? For example, Figure 5A shows that genes with antisense produce more stable sense transcripts than genes without antisense. Is this still true if antisense-containing genes are considered based on nascent transcription and not TIF-seq?

We thank the reviewer for their comment. The antisense-containing genes were indeed defined using a list of annotated transcripts, including those from the above mentioned TIF-seq study. However, this list *also* included the stable unannotated antisense transcripts (SUTs) identified previously by Xu et al., 2009 and the data from Neil et al., 2009., wherein exosome mutants allowed these unstable transcripts (CUTs) to be detected. As such we feel our list is comprehensive and includes both stable and unstable antisense transcripts. We realise now that we failed to report the inclusion of these unstable transcripts in our methods section. We hope this is now clearer in the manuscript.

The use of NET-seq alone to determine whether antisense transcripts are present or absent at a gene is complicated by the fact that NET-seq cannot reliably determine transcription start sites, hence why we supplemented this data with available transcript maps. However, when separating genes into groups of high and low antisense transcription, instead of those with and without antisense transcripts, similar trends are observed. For example, comparing genes with high and low antisense transcription shows the same difference in sense transcript stability as shown in figure 5A:

- Figure 2A: It would be interesting to show the nascent transcription profiles considering the high/low levels of sense transcription and high/low levels of antisense transcription. This information is important to fully appreciate the nucleosome and histone modification profiles shown in Figures 2B and 3.

We thank the reviewer for their comment and have added these figures as Appendix Figure S1.

- Figure 4 and S3: by mathematical modelling, the authors propose that GAL1 gene transcription is decreased in presence of high AS. This prediction should be confirmed by a direct experiment.

We thank the reviewer for their comment. We currently feel this is beyond the scope of available experimental techniques. Assuming an elongation rate of 2kb/min, on a short gene such as the truncated *GAL1*, we would be unable to assess a direct elongation rate by pulse-labelling run-on techniques (GRO-seq, PRO-seq, TT-seq). Given the long pulse labels times currently used for GRO-seq, PRO-seq or 4sU-labelling, if performed on the truncated *GAL1*, the data would reflect effects of degradation as well as initiation and elongation, i.e effectively steady-state. Similarly, NETseq or PolII ChIPseq profiles are defined by both initiation and elongation. Currently we feel that analysis of RNA FISH remains the best experiment to determine this. We have though shown that the relationship between antisense transcription and sense transcription holds at other genes with varying levels of antisense transcription.

- The authors suggest that their observations may reflect a general mechanism. However there is a discrepancy with data already published by Eser et al., 2016. Using 4-ThioU, this paper proposes that in *S. pombe* antisense transcription slows down mRNA synthesis but does not affect mRNA stability.

For these reasons, besides the data from Wang et al. 2002 used in Figure 5A, it may worth also analysing the data from Sun et al. 2013 to confirm that genes with AS transcription produce mRNAs with a longer half-life.

We thank the reviewer for their comments. First we deal with the issue of transcript stability. We have assessed the difference in transcript stability using datasets beyond the data from Wang et al. 2002. Below are shown the distribution of half-lives obtained from Geisberg et al., 2014 and Miller et al., 2011:

Additionally, when we look at the distribution of the *ratio* of RNA-seq to NET-seq reads at genes – a ratio which one would expect to reflect transcript stability - we also see this same difference:

These data have been included in the revised manuscript (Figure EV5). We feel that the use of these four data sources, all of which show the same difference, comprehensively demonstrates the relationship between antisense transcription and sense transcript half-life.

Next, we address the comments about the Eser paper studying *S.pombe* antisense transcription. We do not believe there is a discrepancy with the published data and present our arguments below. We have taken the annotations from Eser et al. (2016) and compared their derived synthesis rates with the given annotations, separating genes into those with no overlapping TU, those overlapping at least one coding TU, those overlapping at least one non-coding TU and those overlapping at least one coding and non-coding TU and find no significant difference in synthesis rates across these 4 classes. While degree of overlap may cause a correlation with synthesis rate as reported in the Eser paper, this is not the case when comparing those with no overlap to those with overlapping antisense or more complex structure, which represents the analysis we have done here. There is an issue with the number of transcripts that overlap with the sense promoter. In the appendix figure S4 the authors show that only ~5% of their overlapping antisense TUs have their 3' end in the 5' UTR of a sense TU. This means that >90% of the antisense TUs analysed in this work do not read into the

promoter of the sense TU, contrary to the form of antisense transcription we look at in our system.

- The GAL1 locus may not be the optimal choice as a model gene since it has a complex profile of ncRNA transcription. Moreover, the upstream GAL10 ncRNA (ncRNA upstream of GAL1) may be more important than the GAL1 antisense since its repression leads to increase of GAL1 expression (Houseley et al., 2008) (which is not the case of GAL1 antisense repression). Thus, GAL1 presents a complex ncRNA pattern and is not solely influenced by its antisense.

Regarding this comment, we refer the reviewer to work, including our own (e.g. Mellor 2016 TIGs), showing that gene rich regions of eukaryotic genes are pervasively transcribed with transcription on both strands and often over the promoter. *GAL1* is not alone in having a complex profile of ncRNA transcription, and in fact is probably a very good model for studying the effects of antisense transcription as it does represent a common situation, with multiple overlapping transcripts at the level of the population of cells. It is worth pointing out that the levels of the upstream *GAL10* lncRNA

do not change in our two constructs – this has been dealt with in previous publications (Murray et al 2012). Moreover, the interpretation of the reviewer of the Houseley experiment is not correct for our growth conditions. Houseley report an effect of the *GAL10* lncRNA on *GAL1* induction and expression only under very artificial and precise growth conditions (almost starvation conditions using very low concentration of glucose and galactose). Moreover, other groups have reported exactly the *opposite* effects of the *GAL10* lncRNA on *GAL1* induction (PLoS Biol. 2013 11:e1001715. doi: 10.1371/journal.pbio.1001715), but again these are minor effects observed using different growth condition from the one used here. To reiterate, we see no effects on loss of the *GAL10* lncRNA on *GAL1* induction kinetics under our standard growth conditions and thus is not an issue.

The authors should confirm their model by experimental analysis of natural genes presenting either high or low levels of AS transcription.

We thank the reviewer for this suggestion. We have done this experiment and have included the data in the main text (Fig 5B) where elongation/export rate is analysed relative to initiation rate for a number of genes with varying levels of antisense transcription. As the initiation rate of transcription is a product of multiple factors, the relative processing rate remains the only factor that can be “anchored” in these endogenous genes with varying antisense. When this is done, there is a clear relationship with levels of antisense transcription, thus validating our model.

Minor points

- Figure 2B : not clear whether the genes with high/low sense are taken among the total number of genes, or among the 1500 AS producing genes ? Do the high/low sense genes really produce no AS ?

The genes with high/low sense are taken from the ~1500 genes with an antisense transcript, as they are the only genes at which an aTSS was available. High/low sense genes are fully capable of producing AS, as shown in fig 2C and 2D.

- Figure 2E : Why analyse 1045 and 1046 genes with low versus high S, and not the same 300 as in the previous figure ? Which genes were chosen in this case ? They have no AS ?

The genes in question are taken from the total pool of yeast genes, split into genes with high and low sense transcription. When we only consider those genes *with* an antisense we see the same lack of correlation between sense transcription and gene compaction. This has been amended in figure 2.

- p.6 : « there is no difference in gene activity », is misleading since it may be understood as transcriptional activity, which is proposed later to be decreased in presence of high AS.

We have corrected this statement in the manuscript.

- In the previous publication (Murray et al., 2015), most of the metagene profiles were normalized to H3, which is not the case here (Figures 3 and S2). Why this difference? In contrast, the ChIP results are normalized to H3 (Figure 6 and S5); in this case the H3 levels should be shown since based on earlier data, they are not the same in the 2 strains expressing high and low GAL1 AS (Murray et al., 2015).

In this manuscript, we did not normalise the data in the metagenes to H3 because we were interested in making a direct comparison between the yeast data and the HeLa data. The HeLa cells data did not have matching H3 levels and so there was no opportunity to normalise the data. We hope this clears up any confusion.

The reviewer is correct, we have normalised the *GAL1*-specific data to H3. The reason is that this is a single locus as opposed to a genome wide analysis and thus we need to normalise the data due to the greater variability in ChIP between experiments at a single locus. We have presented the *rRNA* levels alongside the normalised data for the *GAL1* data.

- Figure 4 : how can one exclude that GAL1 mRNA half-life is not affected by the terminator and the mutations therein in an AS independent way. Where does sense transcription exactly end? How is sense 3' cleavage and polyadenylation affected by the mutations in the terminator?

We showed the location of the 3' end of the corresponding *GAL1* mRNAs with high/low antisense in Murray et al. 2015. There is no difference in the major or minor polyadenylation sites in the two constructs. Regarding the transcript stability, (i) we see genome-wide relationships between antisense and transcript stability using a number of different data sets (ii) the *set3* mutant alters

transcript stability independently of the construct used suggesting that any 3' end effect are not the major cause of the differences transcript stability.

- Figure 4D : At what time-point was sense RNA measured. Why normalize to high AS and not some other internal control?

Sense mRNA was measured at 2 hours galactose. mRNA was chosen to be normalised to high AS levels as the corresponding "wild-type" for this experiment. This was done for each experiment.

- p. 10 : "H3K9Ac may result from but is not causally related to histone turnover". This statement sounds contradictory. If an event results from another event, these events are causally related.

The thank the reviewer for pointing this out. We know that substitution of H3K9 does not influence histone turnover (therefore not causing histone turnover) but that histone turnover is associated with acetylated histones and increased H3K9ac can result from increased turnover. We have reworded the text to clarify this.

- Fig. 6D : the levels of H3K9Ac are higher at the GAL1 gene with high AS and the sense transcription rate is lower compared to GAL1 with low AS (Figures 4 and 5). Does the lower GAL1 sense transcription rate (in GAL1 high AS) contribute to the higher H3K9Ac levels by recruiting less histone deacetylases?

This is an interesting suggestion. The Buratowski lab has proposed that H3K4me2 recruits the Set3 complex to chromatin, although there are other likely recruitment mechanisms. We do not see a change in the levels of H3K4me2 at genes with high or low antisense, although the distribution of the modification is altered (Murray et al 2015). If H3K4me2 is the only way to recruit Set3 to chromatin, we would predict that there should be no difference in recruitment of Set3 in the presence of absence of antisense transcription, however, this needs to be validated experimentally and this is beyond the scope of the current manuscript.

Reviewer #3:

In this manuscript Brown et al. start by describing the chromatin signature associated to antisense (AS) transcription in budding yeast and humans. Interestingly they find an association between antisense expression and decreased chromatin compaction. In paralleled to this, the authors investigate the effect of AS transcription by modeling transcription of a sense gene and fitting their model with data obtained from single-molecule RNA-FISH. In particular they use a system that they have previously developed where the same gene (GAL1) can be expressed with either high or low level of AS transcript. Combining this data with measures of mRNA stability, they identify that although the abundance of mRNA does not change significantly, the general turnover of GAL1 is accelerated in the absence of antisense transcription. Following this initial observation, the authors hypothesize that some of the observed chromatin marks identified in their first part could be involved in this process. By deleting SET3 the authors are able to mimic in the strain with low AS a decreased RNA turnover (similar to the strain with high AS). Using this information, the authors conclude that antisense transcription, by modifying the chromatin status, is able to affect the posttranscriptional life of the mRNA.

Many researchers are trying to dissect the process by which changes in the nucleus can affect the

post-transcriptional life of the RNA. However, we are still far from a clear understanding of this process. I think this work significantly contributes to our understanding of this process. The study of the crosstalk between nucleus and cytoplasm is very challenging due to the high interdependence between all the factors. In this manuscript the authors perform a thoughtful analysis of the GAL1 system. It would be great to expand this study to different loci, or even genome-wide. However, due to the amount of work required for that study, I think it will be out of the scope of the current manuscript. I think this paper is well written and provides insightful information to understand the control of gene expression. However, to improve its clarity and maximize its impact, I have the following questions.

We thank the reviewer for their comments on our work and agree that parts of the manuscript can be clarified and have attempted to do this.

1. The usage of *set3Δ* mimicking the presence of an antisense is very interesting. However, as we know that *set3Δ* can also generate novel cryptic transcripts (Kim et al 2012 PMID 22959268). Does the theoretically low antisense strain produce high antisense after *SET3* deletion? Or is it still behaving as a low antisense strain as expected? A Northern blot or qPCR analysis would clarify this point.

We thank the reviewer for highlighting this. The answer is that a novel cryptic transcript at *GAL1* is *not* the explanation. We show the levels of *GAL1S* and *GAL1AS* transcripts before and after *SET3* deletion by Northern blot in figure 6. There is no difference in the levels of the *GAL1 AS* transcript after deletion of *SET3*. We have made this clearer in the text.

The reviewer is also referred to the responses above concerning the global effects of loss of *SET3*. We show (Figure EV6) that there is no difference at the global level of antisense transcripts, despite *set3* mutants producing some cryptic transcripts at select genes.

2. In Figures 2 and 3 the authors split the pairs sense antisense according to expression levels of sense or antisense transcripts. In some cases it seems that genes with high levels of antisense are very similar to the genes with low level of sense, and vice versa. However, the expression levels of sense and antisense transcripts are not correlated. It would be very useful to have some examples to show the overlap between those different groups (e.g. some Venn diagram depicting the overlap between the 4 groups). Do the shared genes drive the observed chromatin pattern? Or can it be also observed when studying non-overlapping pairs of sense-antisense? Showing some examples for selected panels would be very informative for the overall conclusion.

We thank the reviewer for their comment and have done as requested. We have isolated these four groups of genes – the four different permutations of high/low sense and antisense genes, as is now shown in Appendix Figure S1. As can be seen, there seems to be an additive effect of having high sense *and* high antisense on H3 acetylation, while the two seem to negate each other's effects on H3K36me3 and H3K79me3. We are not sure what is meant by non-overlapping pairs of sense-antisense, as antisense transcripts by definition overlap their paired sense transcript.

3. Previous papers have suggested that sense and antisense transcripts can pair in vivo (eg Wery et al 2016 PMID 26805575). How do the authors discard those that could drive the observed changes in stability? Following the same question, do sense and antisense GAL1 molecules coexist in the same cells? Do the authors have tried to image AS GAL1 using their RNA-FISH approach? Some clarification and extended discussion would be helpful.

The work from the Morillon lab does indeed suggest that the formation of dsRNA (sense antisense pairs) leads to NoGo decay but this is in a mutant background (*xrn1*) when normally unstable transcripts are stabilized artificially. Moreover, this was a relatively mild phenomenon at a few selected loci. Data from the Steinmetz and Knop groups shows that altering levels of 162 natural antisense transcripts in yeast does not influence protein levels produced from the mRNAs, so this is not likely to be a natural phenomenon (Huber 2016). Data from our lab and others (Stutz, Dean) supports the idea that sense and antisense transcripts, when stable, are present at distinct cells in the population (see Nguyen et al 2014 for example). We have not been able to image both the GAL1 AS and S by RNA FISH due to some technical problems.

4. The Fig 7 could be improved by adding some more details or connection lines between the

different players. It is difficult to depict a working model integrating the effect of the AS transcript, increase turnover, and SET3.

We agree with the reviewer that this diagram could have been clearer. We have amended the model along the lines suggested and also represented the data in a different format in Figure 7.

Some typos:

- Page 3: The first citation of Fig 3B, should be 3A.
- Page 33: extra "(" in the citation of Longtine et al.

These have been corrected in the main text.

Thank you again for submitting your work to Molecular Systems Biology. We have now heard back from the reviewer who was asked to evaluate your study. As you will see below, reviewer #2 thinks that most issues have been satisfactorily addressed. S/he raises however one remaining concern, related to the data added in Figure 5C, which we would ask you to address in a revision of the manuscript.

REVIEWER REPORT

Reviewer #2:

In this revised version of the manuscript, the comments of the reviewers have mostly been addressed and the overall message improved, although the mechanism by which Set3 may buffer the effect of antisense transcription is still unclear.

One thing to mention is that the answer given to the question of Reviewer 2 on the relationship between high/low antisense transcription and mRNA stability is not totally satisfying. In their response letter (p. 8), the authors show 3 graphs generated from data published by Geisberg et al. (2014), Miller et al., (2011) and a third graph showing the log(RNA-seq/NET-seq) at genes with high/low AS (reference not mentioned). These 3 graphs have also been included in the extended view Figure 5C of the revised manuscript. However, the 3 figures presented in the rebuttal and in the revised MS do not look the same and the calculated p-values are different, although the groups of genes considered and overall conclusion are the same. The authors should explain the origin of this discrepancy and make sure that they present the correct result.

Another minor remark is that the numbering and order of the figures in the revised manuscript is confusing and makes the reading very laborious. Indeed main figures and expanded views have no numbers, and it is unclear in what order they have been put at the end of the MS.

Regarding the reviewers comment:

One thing to mention is that the answer given to the question of Reviewer 2 on the relationship between high/low antisense transcription and mRNA stability is not totally satisfying. In their response letter (p. 8), the authors show 3 graphs generated from data published by Geisberg et al. (2014), Miller et al., (2011) and a third graph showing the log(RNA-seq/NET-seq) at genes with high/low AS (reference not mentioned). These 3 graphs have also been included in the extended view Figure 5C of the revised manuscript. However, the 3 figures presented in the rebuttal and in the revised MS do not look the same and the calculated p-values are different, although the groups of genes considered and overall conclusion are the same. The authors should explain the origin of this discrepancy and make sure that they present the correct result.

Here we provide an explanation for the discrepancy between the data in the responses to the reviewers and shown in Figure 5EVC (now Figure 3EVC).

The 3 figures presented previously in the response to reviewers differ from those in extended figure 5C (now Figure EV3C) by whether the genes were grouped by level of antisense transcription (high/low) or by the presence or absence of an antisense transcript - hence why the trends are the same but the p-values differ. We apologise for this confusion and have replaced the figures in the response to reviewers with the same figures that are shown in the manuscript to ensure consistency.

Corresponding Author Name: Jane Mellor

Manuscript Number: MSB-17-8007